# AsyncBEV: Cross-modal flow alignment in Asynchronous 3D Object Detection

**Shiming Wang, Holger Caesar, Liangliang Nan, Julian F.P. Kooij**
Delft University of Technology
{s.wang-15, h.caesar, liangliang.nan, j.f.p.kooij}@tudelft.nl

## Abstract

In autonomous driving, multi-modal perception tasks like 3D object detection typically rely on well-synchronized sensors, both at training and inference. However, despite the use of hardware- or software-based synchronization algorithms, perfect synchrony is rarely guaranteed: Sensors may operate at different frequencies, and real-world factors such as network latency, hardware failures, or processing bottlenecks often introduce time offsets between sensors. Such asynchrony degrades perception performance, especially for dynamic objects. To address this challenge, we propose AsyncBEV, a trainable, lightweight, and generic module to improve the robustness of 3D Bird's Eye View (BEV) object detection models against sensor asynchrony. Inspired by scene flow estimation, AsyncBEV predicts a dense 2D flow field directly from asynchronous multi-modal BEV features taking into account the known time offset between these sensor measurements. The predicted feature flow is then used to warp and spatially align the feature maps, which we show can easily be integrated into different current BEV detector architectures (e.g., BEV grid-based and token-based). Extensive experiments demonstrate AsyncBEV improves robustness against both small and large asynchrony between LiDAR or camera sensors in both the token-based CMT and grid-based UniBEV , especially for dynamic objects. We significantly outperform the ego motion compensated CMT and UniBEV baselines, notably by $16.6\%$ and $11.9\%$ NDS on dynamic objects in the worst-case scenario of a $0.5\,s$ time offset. Code is available at https://github.com/tudelft-iv/AsyncBEV.

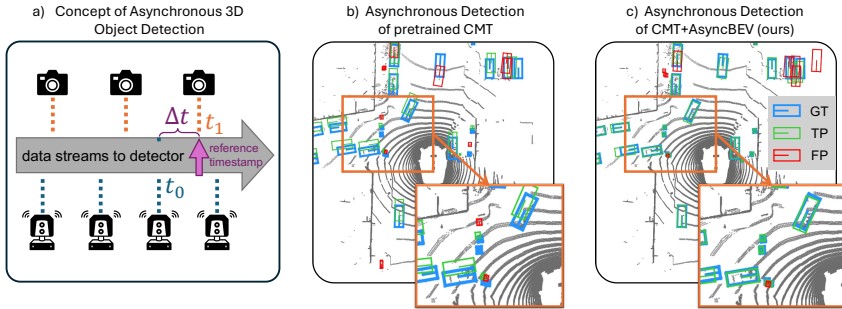

Figure 1: **Multi-modal 3D object detection under sensor asynchrony.** a) shows a general concept of asynchronous 3D object detection. b) shows multi-modal 3D object detection under sensor asynchrony of a pretrained CMT (Yan et al., 2023). Predictions exhibit large translation errors. c) shows our AsyncBEV improves CMT's robustness against sensor asynchrony and mostly corrects the spatial misalignment of predicted boxes.

## 1 Introduction

Autonomous vehicles are typically equipped with multiple sensors such as LiDAR, camera, and radar to support perception tasks (Caesar et al., 2020; de Groot et al., 2025). Temporal synchronization among these sensors is crucial for consistent multi-sensor fusion, and autonomous driving

systems (Sun et al., 2020; Wilson et al., 2021; Palffy et al., 2022) therefore employ various hardware- and software-based synchronization strategies (Sivrikaya & Yener, 2004; Gurumadaiah et al., 2025; Yuan et al., 2022; Faizullin et al., 2022). However, perfect synchronization is hard to guarantee in practice. First, sensors may operate at different sampling frequencies or cannot be triggered synchronously with other sensors (e.g. many radar sensors), hence time offsets between them cannot be avoided. Second, competition for computational resources among different parallel components of the processing pipeline can lead to spurious delays, leading to delayed or dropped frames (Fan et al., 2025; Becker et al., 2020). Finally, common sensor failures, such as sensor crash or even adversarial attacks, may further exacerbate misalignment (Xie et al., 2023; Shahriar et al., 2025).

However, downstream multi-modal 3D object detectors are usually trained on well-synchronized and well-curated datasets, and do not consider the possibility of asynchronous inputs. Figure 1 illustrates this: when one modality is delayed, a detector pretrained on synchronized data produces spatial errors in bounding box predictions, which can compromise safety in autonomous driving.

Ego Motion Compensation (EMC) is widely used in autonomous driving to aggregate the temporal information to improve detection performance (Caesar et al., 2020; Li et al., 2022b; Wang et al., 2023b; Park et al., 2023; Cai et al., 2023; Huang & Huang, 2022; Yin et al., 2021). EMC can compensate the spatial offset between sensor measurements at two time frames arising from the known movement of the ego vehicle, which can align cross-modal features of static objects, but it fails on dynamic objects at longer time intervals. Scene flow methods, on the other hand, estimate the motion of dynamic objects (Zhang et al., 2025a), but they require two point clouds as input and assume fixed time offsets, which limits their applicability to asynchronous multi-modal settings.

To address sensor asynchrony, we propose **AsyncBEV**, a lightweight and generic module that can be seamlessly integrated into diverse architectures of multi-modal 3D object detectors. AsyncBEV predicts the additional dynamic object motion not yet captured by EMC to align the asynchronous sensor data to a reference timestamp. To this end, we define a new task, $\Delta$-BEVFlow estimation, which aims to predict the flow in the BEV space between multi-modal BEV features given the known time offset between the features. The offset-aware design of AsyncBEV facilitates the motion prediction for time offsets that can vary during operation, without significant performance loss in the synchronization case. Since AsyncBEV computes flow from BEV features, it does not make assumptions on the sensor modalities, nor which sensor is considered the reference. To summarize, our main contributions are as follows:

- We introduce AsyncBEV, a trainable and lightweight module that can be integrated into diverse multi-modal 3D object detector frameworks, and improves a detector's robustness against small, moderate, and severe time differences between sensors, especially for dynamic objects. We integrate AsyncBEV in both a token-based and a dense grid-based detector architecture.

- AsyncBEV utilizes $\Delta$-BEVFlow, a novel task that predicts flow in the BEV feature space. Unlike prior work on scene flow, $\Delta$-BEVFlow operates on multi-modal feature maps rather than point clouds, and is conditioned on the known time interval between the sensors.

- We study two different flow formulations: Motion-based, which regresses the flow between BEV features directly, whereas velocity-based flow predicts object velocities, and afterward multiplies these with the known time interval to obtain motion. The velocity-based formulation regularizes the task, resulting in better performance than motion-based flow regression.

## 2 RELATED WORK

**Multi-modal 3D Object Detection.** To leverage the complementary information of different modalities (e.g., LiDAR and cameras), multi-modal 3D object detection has drawn wide attention in both academia and industry (Song et al., 2024b; Wang et al., 2023c; Huang et al., 2022). Since cameras and LiDAR capture data in different coordinate systems, object detectors map the multi-modal features into a shared reference system, usually LiDAR coordinates. They thus have to explicitly or implicitly project multi-view image features into the LiDAR (BEV) space. Grid-based methods (Liu et al., 2023; Cai et al., 2023; Wang et al., 2024; Ge et al., 2023; Li et al., 2022a) explicitly project image features into a BEV grid for unified representations, while token-based methods (Yan et al., 2023; Chen et al., 2022; Li et al., 2022a) rely on object queries to implicitly perform coordinate transformation under the guidance of their positional encodings, which represent the mapping

between each token and its 3D coordinate information to perform the coordinate transformation implicitly. These methods require strict cross-sensor synchronization so that the images and point clouds align in the 3D coordinate system, though 3D object detection under sensor asynchrony remains underexplored.

**Scene Flow Estimation.** Scene flow is the 3D motion field of all points in a scene, describing the per-point motion in 3D space between consecutive point clouds. Recent scene flow estimators deploy feed-forward neural networks (FNN) to achieve high accuracy and real-time inference (Jund et al., 2021; Zhang et al., 2024; 2025b; Kim et al., 2024; Lin et al., 2025; Zhang et al., 2025a). Motion cues of dynamic objects, which scene flow provides, could help re-align information from the asynchronous time frames to a reference time frame. However, typical scene flow methods require two adjacent point clouds and assume a fixed time offset, and are therefore not suited for aligning measurements from different sensor modalities at dynamic time offsets.

**Asynchronous Fusion in Collaborative Perception.** Asynchronous fusion is a crucial problem in collaborative perception, as this task needs to fuse information from multiple agents. But distinct agents and roadside units have independent clocks and are hard to synchronize with each other. CoBEVFlow (Wei et al., 2023) first generates object proposals from distinct agent features and then applies a computation-intensive attention-based module to associate the proposals from different agents. Its flow is not directly learned but calculated based on the associated object proposals. UniV2X (Yu et al., 2025) employs an agent query-based flow predictor to generate the potential motion of the query, which takes the associated agent from multiple frames as input. Nevertheless, these flow calculation methods have two main shortcomings. First, their performance of flow generation heavily relies on the quality of the generated object proposals. However, in multi-modal (on-board) 3D object detection, the detection characteristics for different sensors can differ drastically. For instance, camera detectors usually perform worse than LiDAR detectors. Second, even assuming correct object proposals, in these two works, all agents send homogeneous features from the same modality (LiDAR features in CoBEVFlow (Wei et al., 2023) and image features in UniV2X (Yu et al., 2025)). The distinct features in multi-modal (on-board) 3D object detection pose further challenges in associating object proposals generated from different sensors or predicting query flows with a single MLP module. The effectiveness of their temporal alignment method has not been validated in a multi-modal setup.

**Asynchronous Fusion in On-board Perception.** Asynchronous fusion in on-board perception shares a common concept with temporal feature aggregation, i.e., to align features from another timestamp to the reference timestamp. Early works (Caesar et al., 2020; Huang & Huang, 2022; Park et al., 2023) pioneered the use of EMC to align the static objects, while following works (Li et al., 2022b; Wang et al., 2023b; Cai et al., 2023) further apply attention modules to encode the object motion implicitly. StreamingFlow (Shi et al., 2024) and TimeAlign (Song et al., 2024a) pioneered autonomous driving perception tasks with asynchronous input. However, both works deploy a computation-heavy RNN-based module to predict features at the reference timestamp, taking the long sequence history data as input. StreamingFlow (Shi et al., 2024) takes streaming multi-modal features as a set of observations and treats these features uniformly, which requires a unified feature representation, such as grid-like BEV features, for both modalities. Hayes et al. (2024) leverages the velocity information provided by radars to update the point positions from the asynchronous timestamp to the reference timestamp, which shares the same spirit as scene flow estimation. Fan et al. (2025) proposes directly integrating time features into per-point features to make the detector time-aware. None of the above methods explicitly uses the motion information from multi-modal features. Hence, to address the sensor asynchrony problem, we propose AsyncBEV, a lightweight, generic, interpretable and effective solution, which can work with arbitrary time offsets, asynchrony of different sensors, and distinct 3D object detectors.

## 3 METHODOLOGY

Most multi-modal 3D object detectors follow a similar structure, which we describe here first under the common simplifying assumption that all sensors are synchronous. This paper mainly considers camera and LiDAR fusion, though the same principles can be applied to include radar, too.

As input, the detector receives images $\mathbb{I} \in \mathbb{R}^{V \times H_{\text{img}} \times W_{\text{img}} \times 3}$ from $V$ camera views, and a point cloud $\mathbb{P} \in \mathbb{R}^{(N^{t_0}, 3)}$. First, $\mathbb{I}$ and $\mathbb{P}$ are fed into two separate backbones to generate modal-specific

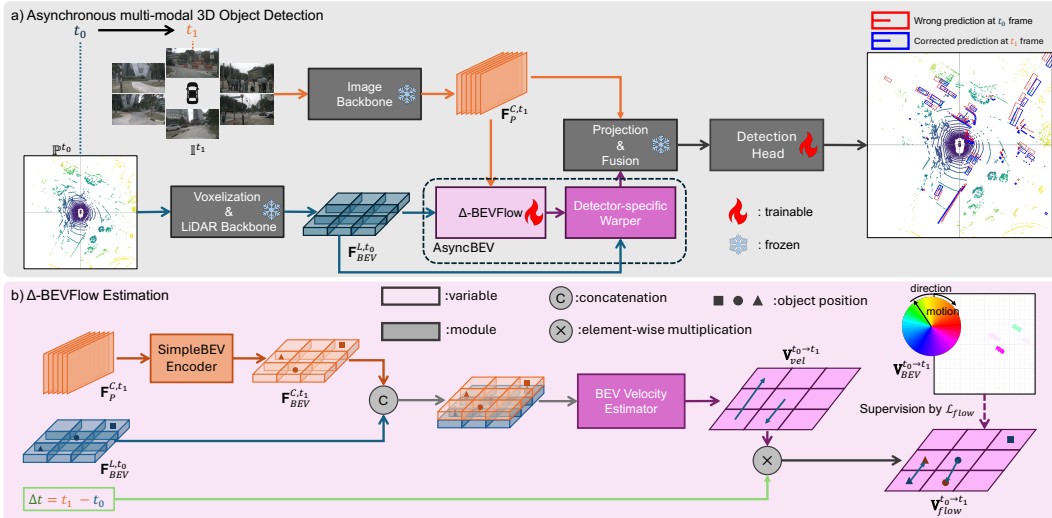

Figure 2: **Overview of AsyncBEV**. a) demonstrates an example pipeline of asynchronous multi-modal 3D object detection. In this setup, images are sampled at the reference timestamp $t_1$, while LiDAR point clouds were obtained at an earlier timestamp $t_0$. b) shows the $\Delta$-BEVFlow estimation, the core module of AsyncBEV. It takes asynchronous BEV features as input and predicts the velocity of each BEV cell. The BEV flow is calculated by multiplying the velocity by $\Delta t$ and supervised by the ground truth dense flow representation. Finally, the predicted flow is used to spatially align the features from $t_0$ to $t_1$ with the detector-specific warper.

features in their native coordinate, resulting in multi-view image features in perspective view $\mathbf{F}_{\mathrm{P}}^{C} \in \mathbb{R}^{V \times H_C \times W_C \times D_C}$ and LiDAR features in bird's eye view $\mathbf{F}_{\mathrm{BEV}}^{L} \in \mathbb{R}^{H_L \times W_L \times D_L}$, where $H_C$, $W_C$, $D_C$ and $H_L$, $W_L$, $D_L$ represent the sizes and dimensions of perspective view (PV) image features and bird's eye view (BEV) LiDAR features, respectively. After getting the multi-modal features in their native coordinate systems $\mathbf{F}_{\mathrm{P}}^{C}$ and $\mathbf{F}_{\mathrm{BEV}}^{L}$, different detectors deploy distinct strategies to map sensor features from sensor-specific coordinates to a shared 3D coordinate space. As LiDAR is already defined in the BEV space, these methods mainly differ in how to process camera features $\mathbf{F}_{\mathrm{P}}^{C}$ to corresponding 3D coordinates $\mathbb{C}_{3D}^{C}$. We shall distinguish two main categories of detectors, and show how to integrate AsyncBEV with each: grid-based and token-based methods.

*Grid-based methods* (Liu et al., 2023; Liang et al., 2022; Wang et al., 2024) learn to project $\mathbf{F}_{\mathrm{P}}^{C}$ into a BEV grid with $\mathbb{C}_{3D}^{C}$, resulting in a dense grid-like camera BEV feature map $\mathbf{F}_{\mathrm{BEV}}^{C}$, similar to the LiDAR feature map $\mathbf{F}_{\mathrm{BEV}}^{L}$.

*Token-based methods* (Yan et al., 2023; Wang et al., 2023a) use a sparse set of object queries, which are defined in the 3D space, to retrieve information from features $\mathbf{F}_{\mathrm{P}}^{C}$ through a set of 3D positional encoding generated from $\mathbb{C}_{3D}^{C}$. This achieves the perspective to BEV projection implicitly.

**Asynchrony.** In practice, sensor input can be asynchronous, thus the timestamps associated with data from each sensor will differ. We assume that one sensor, e.g. the camera, is considered the reference that triggers inference of the multi-modal detector at timestamp $t_1$[1]. From the other sensor modality, e.g. LiDAR, the most up-to-date data from timestamp $t_0$ is taken, which will be delayed w.r.t. the reference ($t_0 < t_1$). In this example, we would refer to the cameras as *the synchronous sensor* and the LiDAR as *the asynchronous sensor*, and $\Delta t = t_1 - t_0$ as the asynchrony time offset. In both token-based and grid-based frameworks, the asynchrony between sensors results in a misalignment between $\mathbb{C}_{3D}^{C,t_1}$ and $\mathbb{C}_{3D}^{L,t_0}$, which can lead to severe detection degradation, especially for important dynamic objects. Note that, hereafter, the sensor timestamp $t_0$ or $t_1$ is indicated using superscripts on the corresponding symbols.

---

[1]In this work, we focus only on cross-modality asynchrony, while asynchrony within multiple cameras is considered out of scope.

### 3.1 ASYNCBEV

We propose **AsyncBEV** to predict the flow (movement) of dynamic objects in the BEV space and then warp the misaligned BEV spatial locations before the detector's fusion. Figure 2 a) demonstrates how AsyncBEV can be integrated in a generic 3D object detection framework.

**Flow estimation.**   A basic step to align sensor coordinate frames and feature maps is *Ego-Motion Compensation* (EMC), which represents the known motion of the ego-vehicle between time $t_0$ and $t_1$, and thus aligns static objects only. In the flow formulation, EMC can be represented as a 2D flow field $\mathbf{M}^{t_0 \to t_1} \in \mathbb{R}^{(N^t, 2)}$ which expresses the relative spatial displacement for each BEV grid cell due to ego-motion. Alternatively, scene flow methods aim to predict non-static motion too as a 3D point-level translation $\mathbb{V}^{t \to t+\Delta t} \in \mathbb{R}^{(N^t, 3)}$ from two subsequent point clouds $\mathbb{P}^t \in \mathbb{R}^{(N^t, 3)}$ and $\mathbb{P}^{t+\Delta t} \in \mathbb{R}^{(N^{t+\Delta t}, 3)}$. Therefore, current scene flow estimators can be generally formulated as a function $\mathbb{V}^{t \to t+\Delta t} = \theta_{\text{SceneFlow}}(\mathbb{P}^t, \mathbb{P}^{t+\Delta t})$. Since existing scene flow methods require two point clouds as input and assume a fixed $\Delta t$, they cannot be directly used to address asynchronous multi-modal 3D object detection where $\Delta t$ can vary continuously.

To handle dynamic objects, AsyncBEV adapts the scene flow concept to a new task, $\Delta$**-BEVFlow**, which should capture the motion missing from EMC. $\Delta$-BEVFlow differs from prior scene flow works in several ways: 1) The flow is explicitly conditioned on an arbitrary time offset $\Delta t$, 2) it works with multi-modal BEV features instead of raw point clouds, 3) we only require 2D flow in BEV space. The task is thus to learn a function:

$$\mathbf{V}_{\text{BEV}}^{t_0 \to t_1} = \theta_{\Delta\text{-BEVFlow}}(\mathbf{F}_{\text{BEV}}^{m_0, t_0}, \mathbf{F}_{\text{BEV}}^{m_1, t_1}, \Delta t), \tag{1}$$

where $\mathbf{V}_{\text{BEV}}^{t_0 \to t_1} \in \mathbb{R}^{(H_{\text{BEV}} \times W_{\text{BEV}}, 2)}$ represents the 2D flow field defined in the BEV space with the size $H_{\text{BEV}} \times W_{\text{BEV}}$, and $\mathbf{F}_{\text{BEV}}^{m_0, t_0}$ and $\mathbf{F}_{\text{BEV}}^{m_1, t_1}$ denote the BEV features of modality $m_0$ and $m_1$ sampled from asynchronous timestamps $t_0$ and $t_1$, respectively. We consider two possible formulations for Equation 1: *motion estimation* and *velocity estimation*.

The *motion estimation (BEV-ME)* formulation directly regresses the dynamic motion from $\Delta t$:

$$\mathbf{E}_{me} = \phi(cat(\mathbf{F}_{\text{BEV}}^{C, t_1}, \mathbf{F}_{\text{BEV}}^{L, t_0}, \Delta t)), \quad \mathbf{V}_{\text{BEV}}^{t_0 \to t_1} = \psi(\mathbf{E}_{me}). \tag{2}$$

Here $\phi : \mathbb{R}^{D_L + D_C + 1} \to \mathbb{R}^{D'}$ is an encoder which compresses the (still misaligned) concatenated BEV features to a lower-dimensional BEV feature map, and $\Delta t$, reducing memory and computational requirements. Following common practice in scene flow works, $\psi$ is a U-Net (Ronneberger et al., 2015) which transforms the encoded BEV features into a flow field. We note (Fan et al., 2025) recently also proposes to add $\Delta t$ as additional input dimensions such that a detection network may learn to compensate for dynamic motion, although without an explicit flow formulation.

We here propose an alternative formulation, *velocity estimation (BEV-VE)*. Instead of regressing from $\Delta t$, this first predicts the velocity of each cell independent of $\Delta t$, and afterwards multiplies predicted velocities with $\Delta t$ to obtain the requested motion, exploiting basic knowledge of the relation between motion, velocity and time. This design regularizes the flow at low $\Delta t$ for near synchronous sensors, since it forces the dynamic motion to become zero as $\Delta t$ approaches zero, and simplifies the learning task:

$$\mathbf{E}_{ve} = \phi(cat(\mathbf{F}_{\text{BEV}}^{C, t_1}, \mathbf{F}_{\text{BEV}}^{L, t_0})), \quad \mathbf{V}_{\text{vel}}^{t_0 \to t_1} = \psi(\mathbf{E}_{ve}), \quad \mathbf{V}_{\text{BEV}}^{t_0 \to t_1} = \mathbf{V}_{\text{vel}}^{t_0 \to t_1} \times \Delta t. \tag{3}$$

As our ablation study will validate, velocity estimation works better than motion estimation, and it will be our default $\Delta$-BEVFlow formulation in the other experiments.

**Lightweight image BEV encoder.**   $\Delta$-BEVFlow requires BEV features from all sensor modalities. Since token-based 3D object detectors do not explicitly build image BEV features, we first generate the BEV features directly from the image features, ensuring AsyncBEV can work with different detection frameworks. We adopt SimpleBEV (Harley et al., 2023) to generate $\mathbf{F}_{\text{BEV}}^{C, t_1}$. SimpleBEV directly projects image features $\mathbf{F}_P^{C, t_1}$ into BEV space with the cameras' extrinsic and intrinsic parameters, avoiding demanding operations such as depth estimation (Liu et al., 2023; Liang et al., 2022; Huang et al., 2021) and deformable attention (Li et al., 2022b; Wang et al., 2024).

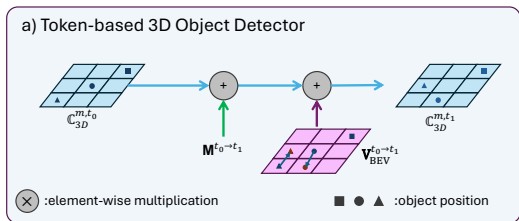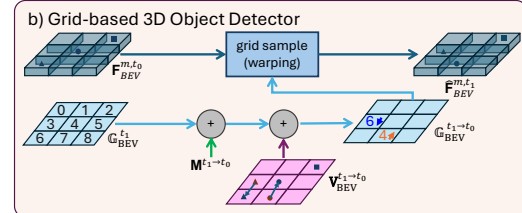

Figure 3: **Detector-specific Warper.** a) Token-based methods rely on 3D coordinates of tokens to provide spatial guidance. Generated $\Delta$-BEVFlow is used to correct the corresponding 3D coordinates of tokens from the asynchronous sensor. b) Grid-based methods project the multi-modal features into predefined BEV grids. Generated $\Delta$-BEVFlow is used to build a look-up table from the reference timestamp to the asynchronous timestamp to warp the asynchronous BEV features to the reference timestamp.

**Detector-specific warper.** Finally, the ego-motion flow $\mathbf{M}^{t_0 \to t_1}$ and $\Delta$-BEVFlow $\mathbf{V}_{\text{BEV}}^{t_0 \to t_1}$ need to be applied to align the features from different sensor modalities. The exact implementation of this step depends on the aforementioned detector architecture, token-based or grid-based.

*Token-based methods* encode spatial locations of tokens (sensor features) in their 3D positional encoding. Therefore, we directly adjust the 3D coordinates $\mathbb{C}_{3D}^{m,t_0}$ of the asynchronous sensor $m \in \{L, C\}$ with the predicted flow $\mathbf{V}_{\text{BEV}}^{t_0 \to t_1}$ to obtain the tokens representing timestamp $t_1$, as shown in Figure 3 a). Specifically, we first apply the predicted dense flow $\mathbf{V}_{\text{BEV}}^{t_0 \to t_1}$ to the position of each sparse token's positional embedding. We do the same operation for the EMC flow, also updating the token positions to compensate for the vehicle ego motion. The final 3D token coordinates $\mathbb{C}_{3D}^{m,t_1}$ are used to generate new positional encodings.

*Grid-based methods* implicitly encode spatial positions in the grid indices. Hence, here we can adopt a *grid sampling* operation to look-up features in the BEV feature map at $t_0$ for each cell in the BEV grid at $t_1$. As shown in figure 3 b), we first define a standard BEV grid $\mathbb{G}_{\text{BEV}}^{t_1}$, representing the 3D space at $t_1$, which should have the same size with BEV features $\mathbf{F}_{\text{BEV}}^{m,t_0}$. Then we simply add the ego-motion flow $\mathbf{M}^{t_1 \to t_0}$ onto $\mathbb{G}_{\text{BEV}}^{t_1}$ to compensate the ego-motion, resulting in $\mathbb{G}_{\text{BEV}}^{t_1,\text{EMC}}$. We further calculate the final look-up table $\mathbb{G}_{\text{BEV}}^{t_1 \to t_0} = \mathbb{G}_{\text{BEV}}^{t_1,\text{EMC}} + \mathbf{V}_{\text{BEV}}^{t_1 \to t_0}$. Note that in this setup, we in fact learn the flow $\mathbf{V}_{flow}^{t_1 \to t_0}$ instead of $\mathbf{V}_{flow}^{t_0 \to t_1}$, as in the case of the token-based models. With $\mathbb{G}_{\text{BEV}}^{t_1 \to t_0}$, we obtain pseudo BEV features of modality $m$ at $t_1$, $\hat{\mathbf{F}}_{\text{BEV}}^{m,t_1}$, with the 'grid_sample' function, i.e., $\hat{\mathbf{F}}_{\text{BEV}}^{m,t_1} = f_{\text{grid\_sample}}(\mathbf{F}_{\text{BEV}}^{m,t_0}, \mathbb{G}_{\text{BEV}}^{t_1 \to t_0})$.

**Loss functions.** AsyncBEV can be trained by back-propagating through an existing frozen object detector framework, using the regular object detection losses. Following the common practice of 3D object detection losses, we also adopt focal loss for classification $L_{\text{cls}}$ and L1 Loss for 3D bounding box regression $L_{\text{reg}}$. A benefit of the explicit flow formulation is that we can (optionally) also directly supervise it to facilitate training. For this, we adopt the flow loss $\mathcal{L}_{\text{flow}}$ from DeFlow (Zhang et al., 2024), using generated ground truth BEVFlow from annotated 3D object bounding boxes, as elaborated in Appendix A.9. This flow loss sums the mean L2-distance between the predicted and ground truth flows for three motion speed categories, i.e. $v \leq 0.4\,m/s$, $0.4\,m/s < v \leq 1\,m/s$ and $v > 1\,m/s$. The overall training objective is then a weighted sum of the three components, $\mathcal{L}_{\text{total}} = \omega_1 \cdot \mathcal{L}_{\text{flow}} + \omega_2 \cdot \mathcal{L}_{\text{cls}} + \omega_3 \cdot \mathcal{L}_{\text{reg}}$, where $\omega_1$, $\omega_2$, and $\omega_3$ are hyperparameters that balance the contributions of the loss terms.

## 4 EXPERIMENTS

**Dataset.** We conduct experiments on the nuScenes (Caesar et al., 2020) dataset. nuScenes is a large-scale autonomous driving dataset collected in Boston and Singapore. It has 1000 scenes in total and is divided into train/val/test sets with 750/150/150 scenes. nuScenes contains multi-modal data from 6 cameras, 1 LiDAR, and 5 radars. In this work, we mainly focus on LiDAR and

| Method | | All Objects | | | Static Objects | | | Dynamic Objects | | | FPS ↑ |
|---|---|---|---|---|---|---|---|---|---|---|---|
| | | 0 s | 0.25 s | 0.5 s | 0 s | 0.25 s | 0.5 s | 0 s | 0.25 s | 0.5 s | |
| CMT | NDS ↑ | 72.9 | 49.6 | 43.2 | 69.0 | 48.7 | 44.2 | 47.5 | 32.2 | 26.1 | 6.7 |
| | mAP ↑ | 70.3 | 35.9 | 24.8 | 61.0 | 32.2 | 22.8 | 36.8 | 16.4 | 9.0 | |
| CMT + EMC | NDS ↑ | 72.9 | 66.8 | 63.3 | 69.0 | 68.3 | 67.5 | 47.5 | 34.5 | 26.8 | 6.7 |
| | mAP ↑ | 70.3 | 60.6 | 54.1 | 61.0 | 59.8 | 58.6 | 36.8 | 21.1 | 11.9 | |
| CMT + DA | NDS ↑ | 71.5 ↓1.4 | 70.2 ↑3.4 | 67.6 ↑4.3 | 68.2 ↓0.8 | 67.2 ↓1.1 | 65.4 ↓2.1 | 45.8 ↓1.7 | 44.9 ↑10.4 | 41.5 ↑14.7 | 6.7 |
| | mAP ↑ | 68.4 ↓1.9 | 66.7 ↑6.1 | 63.0 ↑8.9 | 60.2 ↓0.8 | 58.7 ↓1.1 | 56.0 ↓2.6 | 34.5 ↓2.3 | 33.8 ↑12.7 | 30.0 ↑18.1 | |
| Fan et al. (2025) | NDS ↑ | 70.6 ↓2.3 | 70.0 ↑3.2 | 66.2 ↑2.9 | 67.7 ↓1.3 | 67.1 ↓1.2 | 64.7 ↓2.8 | 43.5 ↓4.0 | 43.9 ↑9.4 | 37.9 ↑11.1 | 6.7 |
| | mAP ↑ | 67.9 ↓2.4 | 66.8 ↑6.2 | 61.3 ↑7.2 | 59.1 ↓1.9 | 58.4 ↓1.4 | 54.8 ↓3.8 | 34.7 ↓2.1 | 33.1 ↑12.0 | 27.1 ↑15.2 | |
| CMT+AsyncBEV | NDS ↑ | 72.5 ↓0.4 | 71.5 ↑4.7 | 70.0 ↑6.7 | 68.7 ↓0.3 | 68.3 | 67.6 ↑0.1 | 47.1 ↓0.4 | 46.1 ↑11.6 | 43.4 ↑16.6 | 6.3 |
| | mAP ↑ | 69.8 ↓0.5 | 68.3 ↑7.7 | 66.4 ↑12.3 | 60.7 ↓0.3 | 60.0 ↑0.2 | 59.1 ↑0.5 | 36.8 | 35.4 ↑14.3 | 32.3 ↑20.4 | |
| CMT+AsyncBEV (FD) | NDS ↑ | 73.0 ↑0.1 | 71.4 ↑4.6 | 68.7 ↑5.4 | 69.2 ↑0.2 | 68.2 ↓0.1 | 67.0 ↓0.5 | 47.7 ↑0.2 | 45.0 ↑10.5 | 39.6 ↑12.8 | 6.3 |
| | mAP ↑ | 70.5 ↑0.2 | 68.1 ↑7.5 | 64.4 ↑10.3 | 61.4 ↑0.4 | 59.9 ↑0.1 | 58.3 ↓0.3 | 36.9 ↑0.1 | 34.1 ↑13.0 | 27.8 ↑15.9 | |
| UniBEV | NDS ↑ | 66.7 | 45.7 | 39.7 | 63.5 | 45.3 | 41.2 | 42.4 | 29.6 | 25.3 | 2.8 |
| | mAP ↑ | 62.0 | 32.1 | 22.2 | 52.2 | 27.6 | 19.0 | 32.9 | 15.0 | 8.9 | |
| UniBEV+EMC | NDS ↑ | 66.7 | 61.7 | 58.9 | 63.5 | 62.9 | 62.2 | 42.4 | 31.2 | 25.9 | 2.8 |
| | mAP ↑ | 62.0 | 53.8 | 48.2 | 52.2 | 51.1 | 50.0 | 32.9 | 21.1 | 11.4 | |
| UniBEV + DA | NDS ↑ | 60.2 ↓6.5 | 58.3 ↓3.4 | 54.5 ↓4.4 | 58.5 ↓5.0 | 56.8 ↓6.1 | 53.8 ↓8.4 | 36.6 ↓5.8 | 36.0 ↑4.8 | 33.6 ↑7.7 | 2.8 |
| | mAP ↑ | 53.6 ↓8.4 | 50.9 ↓2.9 | 45.9 ↓2.3 | 45.2 ↓7.0 | 42.9 ↓8.2 | 38.3 ↓11.7 | 24.9 ↓8.0 | 24.0 ↑2.9 | 20.4 ↑9.0 | |
| StreamingFlow (Shi et al., 2024) | NDS ↑ | 64.2 ↓2.5 | 58.8 ↓2.9 | 54.1 ↓4.8 | 61.8 ↓1.7 | 56.8 ↓6.1 | 53.1 ↓9.1 | 41.3 ↓1.1 | 38.5 ↑7.3 | 34.4 ↑8.5 | 1.0 |
| | mAP ↑ | 60.3 ↓1.7 | 53.0 ↓0.8 | 47.4 ↓0.8 | 49.9 ↓2.3 | 43.2 ↓7.9 | 37.3 ↓12.7 | 31.5 ↓1.4 | 27.4 ↑6.3 | 21.6 ↑10.2 | |
| UniBEV+AsyncBEV | NDS ↑ | 65.7 ↓1.0 | 64.8 ↑3.1 | 63.3 ↑4.4 | 62.7 ↓0.8 | 62.0 ↓0.9 | 61.5 ↓0.7 | 41.0 ↓1.4 | 40.2 ↑9.0 | 37.8 ↑11.9 | 2.7 |
| | mAP ↑ | 60.8 ↓1.2 | 59.4 ↑5.6 | 57.1 ↑8.9 | 50.8 ↓1.4 | 49.7 ↓1.4 | 49.1 ↓0.9 | 31.0 ↓1.9 | 29.8 ↑8.7 | 26.6 ↑15.2 | |
| UniBEV+AsyncBEV (FD) | NDS ↑ | 66.7 | 65.2 ↑3.5 | 62.1 ↑3.2 | 63.5 | 62.9 | 62.0 ↓0.2 | 42.5 ↑0.1 | 39.6 ↑8.4 | 33.8 ↑7.9 | 2.7 |
| | mAP ↑ | 62.1 ↑0.1 | 59.6 ↑5.8 | 54.7 ↑6.5 | 52.3 ↑0.1 | 51.2 ↑0.1 | 49.9 ↓0.1 | 32.9 | 29.1 ↑8.0 | 20.7 ↑9.3 | |

Table 1: **Evaluation on the nuScenes val set of asynchronous multi modal 3D object detection** (**Best**, Second Best). LiDAR is asynchronous. AsyncBEV is only trained once with each model and evaluated on different time offsets. FD: freeze the whole detector and only train AsyncBEV. Performance deltas are measured relative to the EMC baseline. Inference speed in Frames Per Second (FPS) is measured with a single A100 GPU.

camera fusion for 3D object detection. Following common practice in nuScenes, we report both the nuScenes Detection Score (NDS) and mean Average Precision (mAP) as metrics. To understand the motion compensation performance of our approach, we further report NDS and mAP for dynamic and static objects. We adopt $0.2\,m/s$ as a threshold to distinguish dynamic and static objects in both predictions and ground truth, then calculate the metrics respectively.

**Baselines.** We integrate AsyncBEV in both the CMT (Yan et al., 2023) and UniBEV (Wang et al., 2024) architectures, which are representative for token-based and grid-based multi-modal 3D object detectors. Compared to its original end-to-end training schema, we retrained UniBEV with a two-stage training protocol to accelerate the training process. In the default setting, we use frozen pretrained backbones and finetune only the AsyncBEV module and the detection heads unless otherwise specified. We also train a variant *AsyncBEV (FD)* of each architecture with frozen detection heads, so only the AsyncBEV module is trained. We adopt a learning rate of $2 \times 10^{-5}$ for finetuning in both architectures. Other hyperparameters are kept the same as their original training setups. For both CMT and UniBEV architectures, we also test an EMC-only variation without AsyncBEV, which has no additional learnable components. Data augmentation (DA) is another simple baseline for both networks, which directly finetunes the vanilla detectors with the same asynchronous training scheme as AsyncBEV.

Furthermore, we also deploy the two asynchronous fusion methods Fan et al. (2025) and StreamingFlow (Shi et al., 2024) as baselines. Fan et al. (2025) proposes a temporal information propagation method for point clouds. Modern LiDAR encoders usually accumulate multiple historic sweeps of point clouds with the latest one as input point cloud, and append their time offsets relative to the latest one as an additional temporal feature alongside its geometric dimensions. Following Fan et al. (2025), we simply replace the original temporal feature with the asynchronous time offsets $\Delta t$ measured relative to the reference timestamp. We integrate Fan et al. (2025) with CMT for asynchronous LiDAR. Please note, Fan et al. (2025) can only be applied to the point cloud input, as images usually have no time dimension. A reasonable extension is to concatenate the $\Delta t$ to the intermediate features, which is similar to our motion-based flow estimation variant. StreamingFlow utilizes a GRU-ODE to model the temporal dynamics among grid-like features from different timestamps. To adapt StreamingFlow to 3D object detection task, we therefore integrate the GRU-ODE module into our grid-based baseline, UniBEV. For all the trainable baselines, we also adopt the same training

scheme with AsyncBEV, i.e., keep the pretrained backbone frozen and finetune the detection head and other components.

**Asynchronous training setup.** While our method is generic to any dataset, we follow nuScenes conventions here for simplicity. As is common practice, we train and evaluate our method on annotated nuScenes keyframes at 2 Hz. Considering the keyframe as the reference timestamp, one sensor is synchronous while the other sensor is asynchronous. Since the nuScenes dataset is artificially curated to discard scenes with bad synchronization, we cannot use the natural data, but have to impute asynchronous data by using older sensor data for the asynchronous sensor. During training we vary the time offset $\Delta t$ uniformly at random between $0\,s$ and a maximum of $0.5\,s$. We consider $0.5\,s$ the worst-case scenario for our robustness test, which could be uncommon but high-impact. For the asynchronous sensor we use the sensor data that precedes the reference timestamp by the specified time offset. The cameras have a frequency of 12 Hz and the LiDAR operates at 20 Hz. Since we cannot realistically infer the sensor data at an arbitrary point in time, we instead use the closest asynchronous sensor data to the specified time offset. Therefore we cannot evaluate arbitrarily small time offsets, but only multiples of the sensor period ($83\,ms$ for camera and $50\,ms$ for LiDAR). nuScenes uses 6 cameras which are triggered when the LiDAR scans across the center of each camera's Field of View (FoV). Therefore each camera has a different timestamp. We treat all cameras the same way and assign them the timestamp of the back-left camera, which is closest to the reference timestamp of the corresponding LiDAR sweep.

### 4.1 ASYNCHRONOUS MULTI-MODAL 3D OBJECT DETECTION

Table 1 presents the main evaluation results for large time intervals (severe asynchrony), using the camera as the reference time frame while LiDAR is asynchronous with it. A similar table where LiDAR is synchronous and cameras asynchronous can be found in Appendix A.1, demonstrating the generality of AsyncBEV. Since both setups show similar trends, we focus here on the camera as the reference setup only. Figure 4 demonstrates the NDS for All Objects, and for Dynamic Objects only, at lower time offset ranges, representing more common less severe cases. A similar figure for mAP is shown in Appendix A.6. Note that the last column of Table 1 showcases the FPS of each method, which confirms our AsyncBEV only introduces marginal latency to the vanilla models.

**Vanilla Models.** When the asynchrony between images and point clouds increases to $0.5\,s$, the performance of both 3D object detectors, CMT and UniBEV, decreases dramatically, confirming sensor asynchrony leads to performance degradation if not properly accounted for. Especially, the performance of Dynamic Objects declines significantly from $47.5\%$ NDS to $26.1\%$ for CMT and $42.4\%$ NDS to $25.3\%$ for UniBEV with $0.5\,s$ time offset. Even with a mild sensor asynchrony with $50\,ms\,(0.05\,s)$ time interval, its performance for All Objects and Dynamic Objects drops substantially, as shown in Figure 4.

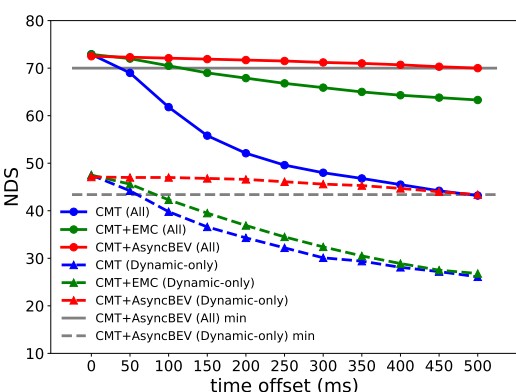

Figure 4: **Performance** of different methods with increasing time offsets under LiDAR asynchrony.

**Ego Motion Compensation (EMC) only.** Table 1 confirms only performing EMC already significantly improves the NDS with $0.5\,s$ asynchrony for CMT and UniBEV. Notably, the performance of static objects at $0.5\,s$ asynchrony improves by $23.3\%$ NDS and to $67.5\%$ for CMT and by $21.0\%$ NDS to $62.2\%$, which reaches a comparable performance of static objects in the synchronous case with $69.0\%$ and $63.5\%$ for CMT and UniBEV. However, EMC does not bring a performance improvement for dynamic objects, as expected. Similar observations can be made in Figure 4: Though the performance of EMC improves the performance for All Objects over the vanilla CMT across all time offsets, its performance for Dynamic Objects barely increases.

**Data Augmentation.** Finetuning the vanilla model with asynchronous inputs as a data augmentation (DA) strategy also improves robustness to sensor asynchrony, but its effectiveness differs between CMT and UniBEV. Compared to CMT+EMC, the performance of CMT+DA in the synchronous case declines by $1.4\%$ NDS but increases in asynchronous scenarios. In contrast, UniBEV+DA underperforms UniBEV+EMC in both cases. Notably, DA consistently improves the performance for Dynamic Objects of both models over the EMC baselines.

**AsyncBEV.** In Table 1, the CMT variants overall outperform UniBEV variants, but AsyncBEV helps improve asynchrony robustness for both types of architectures. At a time offset of $0.5\ s$, CMT+AsyncBEV and UniBEV+AsyncBEV reach $70.0\%$ and $63.3\%$ NDS respectively, surpassing their EMC counterparts with a margin of $6.7\%$ NDS and $12.3\%$ mAP. Notably, both CMT+AsyncBEV and UniBEV+AsyncBEV achieve strong performance for Dynamic Objects, namely $43.4\%$ and $37.8\%$ NDS, which significantly outperforms their counterparts with EMC with a margin of $16.6\%$ and $11.9\%$ NDS, and the vanilla models with a considerable margin of $17.3\%$. For static objects, CMT+AyncBEV performs comparably to CMT+EMC, as expected.

We also observe that the AsyncBEV can introduce a slight performance degradation in the synchronized case ($0\ s$) for both models, in its default setting where we fine-tune the detector head. NDS for all objects shows a modest drop of $0.4\%$ for CMT and $1.0\%$ for UniBEV. However, AsyncBEV (FD) maintains the performance of the vanilla models in this synchronized setting. In this variant all original detector components are frozen, and the velocity-based formulation ensures that for $\Delta t = 0\ s$ no motion is predicted, thus all BEV features remain unaffected. Still, AsyncBEV (FD) underperforms compared to default AsyncBEV on Dynamic Objects in asynchronous cases, due to fewer trainable parameters. Fine-tuning the detector head when training AsyncBEV thus controls a trade-off between these cases. Given the modest effect on synchronous cases and strong improvements on asynchronous cases, we keep AsyncBEV with fine-tuning the detector heads as the default.

Figure 4 confirms AsyncBEV improves the robustness of the vanilla CMT also for smaller time intervals, showing flatter curves for both categories. Notably, the performance for All Objects of CMT+AsyncBEV in an extreme sensor asynchrony case, i.e., $0.5\ s$ time offset, outperforms vanilla CMT with a mild $50\ ms$ time offset.

**Other Baselines.** Fan et al. (2025) propagates temporal features, which expects the network to align the features implicitly according to the given asynchronous time offset. As shown in Table 1, this strategy mitigates some of the misalignment for higher time offsets compared to the EMC baseline. On the other hand, it also exhibits a significant drop in performance for the $0\ s$ case compared to vanilla CMT. Overall its performance is inferior to our proposed AsyncBEV for all time offsets, with the performance gap widening as the time offset increases.

StreamingFlow takes a sequence of past images or LiDAR frames as input, and predicts features for a reference timestamp. Despite its performance in synchronous scenarios drops $2.5\%$ NDS, StreamingFlow improves performance for asynchronous scenarios compared to the vanilla UniBEV. Nevertheless, in both scenarios StreamingFlow still underperforms our AsyncBEV, which only uses the latest asynchronous observations as input. Furthermore, due to the need to process every observation in the input sequence with the GRU-ODE module, StreamingFlow operates significantly slower than AsyncBEV in terms of FPS.

**Qualitative Results.** A benefit of our flow prediction approach is that such visualizations improve the interpretability of AsyncBEV. Figure 5 showcases the qualitative results of CMT and CMT+AsyncBEV under the extreme LiDAR asynchrony case with a $0.5\ s$ time offset. In Figure 5 a), the object in the highlight box cannot be detected by the original CMT due to large spatial misalignment in the asynchronous sensor information. Predicted $\Delta$-BEVFlow in Figure 5 c) aligns with the GT flow shown in Figure 5 d) and matches the spatial offsets between the predicted box and the ground truth box. Thanks to the motion cue provided by $\Delta$-BEVFlow, CMT+AsyncBEV successfully detected the object with a small translation error, as shown in Figure 5 b).

To sum up, AsyncBEV brings significant performance gains for 3D object detectors under both severe and light sensor asynchrony, especially for dynamic objects. The approach is generic and lightweight, improving detector robustness for arbitrary time offsets during operation, supporting different sensor modalities, and distinct 3D object detection architectures.

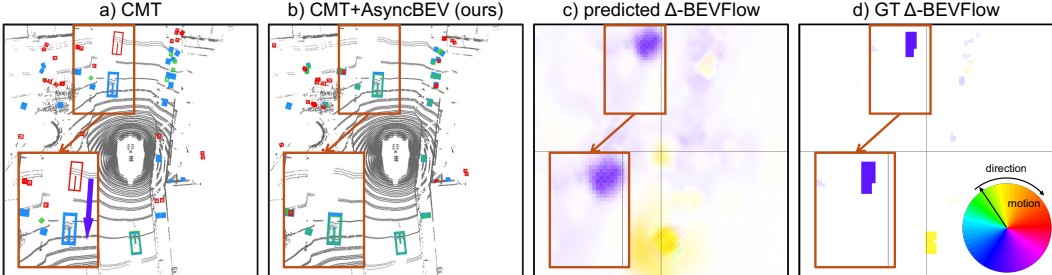

Figure 5: **Qualitative Results of AsyncBEV.** The left two columns demonstrate the 3D object detection performance of CMT and CMT+AsyncBEV under $0.5s$ LiDAR asynchrony. The right two columns show the output of our predicted $\Delta$-BEVFlow and the ground truth $\Delta$-BEVFlow.

## 4.2 ABLATION STUDY OF $\Delta$-BEVFLOW STRATEGY

In this ablation study, we focus on UniBEV only, as performance trends on UniBEV and CMT are generally similar. Table 2 compares the motion-based formulation to the (default) velocity-based formulation of $\Delta$-BEVFlow, and the impact of exposing the model to varying delays during training. The orig-

| $\Delta$-BEVFlow design | | | All Objects ($0\,s$) | | All Objects ($0.5\,s$) | |
|---|---|---|---|---|---|---|
| **EMC+DA** | **Motion** | **Velocity** | NDS | mAP | NDS | mAP |
| ✗ | ✗ | ✗ | **66.7** | **62.0** | 39.7 | 22.2 |
| ✓ | ✗ | ✗ | 63.1 ↓3.6 | 57.6 ↓4.4 | 61.1 ↑21.4 | 54.9 ↑32.7 |
| ✓ | ✓ | ✗ | 64.2 ↓2.5 | 58.6 ↓3.4 | 62.5 ↑22.8 | 56.1 ↑33.9 |
| ✓ | ✗ | ✓ | 65.7 ↓1.0 | 60.8 ↓1.2 | **63.3** ↑23.6 | **57.1** ↑34.9 |

Table 2: **Ablation study of $\Delta$-BEVFlow design.** This table showcases the core design choice for the new task $\Delta$-BEVFlow. **EMC+DA**: EMC + asynchronous data augmentation training. **Motion**: BEV-ME. **Velocity**: BEV-VE, which is our default design.

inal UniBEV performs well in a synchronized case, but fails catastrophically when a serious asynchrony occurs. Based on UniBEV+EMC, we finetune UniBEV with the asynchronous data (**EMC + DA**), which brings significant improvement for the asynchronous case, but also degrades the performance in the synchronized case. **Motion**-based flow estimation (BEV-ME) helps compensate for the dynamic objects. **Velocity**-based flow estimation (BEV-VE) still further prevents the performance drop in the synchronization case, and strengthens the results under extreme sensor asynchrony, validating our choice for this formulation as default.

## 5 CONCLUSIONS

We have proposed AsyncBEV, a lightweight and generic module to improve the robustness of a multi-modal 3D object detector against sensor asynchrony. Generalizing the scene flow concept, we first propose a novel task, $\Delta$-BEVFlow, which predicts the flow in the BEV space given an arbitrary time offset for multi-modal BEV features. The predicted flow can then be applied to spatially align the feature maps of the asynchronous sensor to the reference frame. Our experiments on nuScenes show AsyncBEV significantly improves robustness under sensor asynchrony for both token-based and grid-based detectors, with only little computational overhead. Under an extreme LiDAR asynchrony case with a $0.5\,s$ time offset, AsyncBEV improves the performance for All Objects of vanilla methods with a considerable margin of $26.8\%$ NDS on token-based CMT and $23.6\%$ NDS on grid-based UniBEV. Particularly, AsyncBEV outperforms EMC for Dynamic Objects with $16.6\%$ NDS on CMT and $11.9\%$ NDS on UniBEV. Similar trends are observed for smaller time offsets. We also show a velocity-based formulation for the $\Delta$-BEVFlow estimator helps regularize the predicted motion of dynamic objects over directly predicting the motion.

**Limitations and Future Work.** In this work, we assume that one sensor is always synchronized with the reference frame, while the other may have time offsets. However, in the real autonomous driving application, there can be more than two sensors, of which multiple can be asynchronous at once. Please refer to Appendix A.7 for more details. In the future, we will develop a full-streaming framework to always fuse the most up-to-date data from multiple sensors, including radar.

ACKNOWLEDGMENTS

This work was supported by the 3D Urban Understanding (3DUU) Lab funded by the TU Delft AI Initiative.

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

# A APPENDIX

## A.1 MULTI-MODAL 3D OBJECT DETECTION WITH ASYNCHRONOUS CAMERAS

| Method | | All Objects | | | Static Objects | | | Dynamic Objects | | |
|---|---|---|---|---|---|---|---|---|---|---|
| | | 0 s | 0.25 s | 0.5 s | 0 s | 0.25 s | 0.5 s | 0 s | 0.25 s | 0.5 s |
| CMT | NDS ↑ | **72.9** | 70.7 | 67.5 | **69.0** | 66.9 | 63.8 | 47.5 | 46.9 | 45.8 |
| | mAP ↑ | **70.3** | 66.7 | 61.2 | **61.0** | 57.9 | 52.4 | 36.8 | 36.0 | 34.2 |
| CMT + EMC | NDS ↑ | **72.9** | 71.4 | 68.8 | **69.0** | 67.6 | 65.1 | 47.5 | 46.9 | 45.9 |
| | mAP ↑ | **70.3** | 67.8 | 63.5 | **61.0** | 58.9 | 54.9 | 36.8 | 36.0 | 34.1 |
| CMT+AsyncBEV | NDS ↑ | 72.7 ↓0.2 | **72.7** ↑1.3 | **72.3** ↑3.5 | 68.8 ↓0.2 | **68.8** ↑1.2 | **68.4** ↑3.3 | **48.1** ↑0.6 | **48.1** ↑1.2 | **48.0** ↑2.1 |
| | mAP ↑ | 69.8 ↓0.5 | **69.7** ↑1.9 | **69.1** ↑5.6 | 60.8 ↓0.3 | **60.6** ↑1.7 | **60.2** ↑5.3 | **37.0** ↑0.1 | **36.9** ↑0.9 | **36.7** ↑2.6 |
| UniBEV | NDS ↑ | **66.7** | 64.8 | 62.7 | **63.5** | 61.9 | 60.0 | **42.4** | 41.8 | 40.9 |
| | mAP ↑ | **62.0** | 59.3 | 56.1 | **52.2** | 50.1 | 47.1 | **32.9** | 32.3 | 30.6 |
| UniBEV+EMC | NDS ↑ | **66.7** | 66.5 | 66.2 | **63.5** | 63.5 | 63.2 | **42.4** | 41.9 | 41.0 |
| | mAP ↑ | **62.0** | 61.4 | 60.8 | **52.2** | 52.0 | 51.6 | **32.9** | 32.2 | 30.9 |
| UniBEV+AsyncBEV | NDS ↑ | 66.9 ↑0.2 | **66.8** ↑0.3 | **66.7** ↑0.5 | 63.4 ↓0.1 | 63.3 ↓0.2 | 63.2 | 42.0 ↓0.4 | **42.1** ↑0.2 | **42.2** ↑1.2 |
| | mAP ↑ | **62.0** | **61.8** ↑0.4 | **61.5** ↑0.7 | 51.7 ↓0.5 | 51.6 ↓0.4 | 51.5 ↓0.4 | 32.3 ↓0.6 | **32.3** ↑0.1 | **32.3** ↑1.4 |

Table III: Evaluation on the nuScenes **val** set of asynchronous multi modal 3D object detection. (**Best**, Second Best). Camera is asynchronous. Trained with one model and evaluated on different time offsets. Performance deltas are measured relative to the EMC baseline.

Table III demonstrates an opposite asynchrony, i.e., LiDAR is on the reference timestamp, but cameras are asynchronous. Because the dominating sensor, LiDAR, is still functioning, the performance degradation is not as significant as in Table 1. We can still observe that NDS for All Objects decreases $5.4\%$ to $67.5\%$ for CMT and $5.0\%$ to $62.7\%$ for UniBEV with a $0.5\,s$ asynchronous time offset, respectively. AsyncBEV improves the performance of CMT and UniBEV to $72.3\%$ and $66.7\%$ NDS, respectively. Table III validates the generality of our proposed AsyncBEV.

## A.2 COMPARISON WITH THE SENSOR-FALLBACK STRATEGY

| Method | All Objects | | Static Objects | | Dynamic Objects | |
|---|---|---|---|---|---|---|
| | NDS ↑ | mAP ↑ | NDS ↑ | mAP ↑ | NDS ↑ | mAP ↑ |
| CMT (0.5 s L) | 43.2 | 25.8 | 44.2 | 22.8 | 26.1 | 9.0 |
| CMT_C | 44.7 ↑1.5 | 38.3 ↑12.5 | 47.4 ↑3.2 | 29.5 ↑6.7 | 28.3 ↑2.2 | 14.7 ↑5.7 |
| CMT+AsyncBEV (0.5 s L) | 70.0 ↑25.3 | 66.4 ↑28.1 | 67.6 ↑20.2 | 59.1 ↑29.6 | 43.4 ↑15.1 | 32.3 ↑17.6 |
| UniBEV (0.5 s L) | 39.7 | 22.2 | 41.2 | 19.0 | 25.3 | 8.9 |
| UniBEV_C | 42.6 ↑2.9 | 35.4 ↑13.2 | 46.1 ↑4.9 | 27.1 ↑8.1 | 28.2 ↑2.9 | 13.1 ↑4.2 |
| UniBEV+AsyncBEV (0.5 s L) | 63.3 ↑20.7 | 57.1 ↑21.7 | 61.5 ↑15.4 | 49.1 ↑22.0 | 37.8 ↑9.6 | 26.6 ↑13.5 |
| CMT (0.5 s C) | 67.5 | 61.2 | 63.8 | 52.4 | 45.8 | 34.2 |
| CMT_L | 68.1 ↑0.6 | 61.7 ↑0.5 | 64.1 ↑0.3 | 53.0 ↑0.6 | 46.7 ↑0.9 | 34.5 ↑0.3 |
| CMT+AsyncBEV (0.5 s C) | 72.3 ↑4.2 | 69.1 ↑7.4 | 68.4 ↑4.3 | 60.2 ↑7.2 | 48.0 ↑1.3 | 36.7 ↑2.2 |
| UniBEV (0.5 s C) | 62.7 | 56.1 | 60.0 | 47.1 | 40.9 | 30.6 |
| UniBEV_L | 63.7 ↑1.0 | 56.1 | 60.5 ↑0.5 | 47.7 ↑0.6 | 42.2 ↑1.3 | 31.7 ↑1.1 |
| UniBEV+AsyncBEV (0.5 s C) | 66.7 ↑3.0 | 61.5 ↑5.4 | 63.2 ↑2.7 | 51.5 ↑3.8 | 42.2 | 32.3 ↑0.6 |

Table IV: Performance of single-modality performance of CMT and UniBEV.

In our $0.25\,s$ and $0.5\,s$ delay setups, discarding stale data ($> 100\,ms$) is equivalent to single-modality inference. Table IV compares our AsyncBEV to single-modality. As shown in Table IV, when LiDAR is delayed and a sensor fallback strategy is adopted, camera-only CMT_C achieves $44.7\%$ NDS while camera-only UniBEV_C performs at $42.6\%$ NDS. Both outperform the vanilla CMT ($43.2\%$ NDS) and UniBEV ($39.7\%$ NDS) under the $0.5\,s$ LiDAR asynchrony. This shows that the sensor-fallback strategy can serve as a backup option in cases of severe sensor asynchrony, even though the performance gains remain minor. However, our CMT+AsyncBEV achieves $70.0\%$ NDS, and UniBEV achieves $63.3\%$ NDS even with an extreme $0.5\,s$ time offset for LiDAR, outperforming the single-modality performance significantly.

We make similar observations for the camera-delayed scenario. With a $0.5\,s$ time offset for cameras, CMT+AsyncBEV achieves 72.3% NDS, while UniBEV+AsyncBEV achieves 66.7% NDS, both surpassing the LiDAR-only performance of CMT_L (68.1% NDS) and UniBEV_L (63.7% NDS). This shows that the asynchronous modality still contains rich information for perception tasks. In conclusion, effectively leveraging temporally misaligned information is a better option than completely dropping them.

## A.3 MORE QUALITATIVE RESULTS

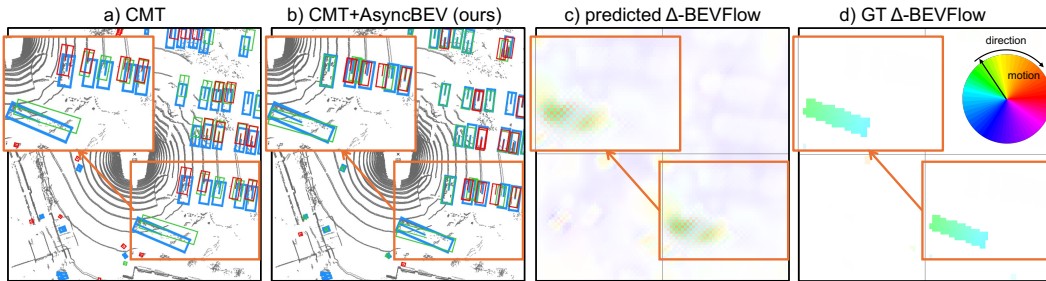

Figure VI: **Qualitative Results of AsyncBEV.** The left two columns demonstrate the 3D object detection performance of CMT and CMT+ AsyncBEV under LiDAR asynchrony with $0.5\,s$ offset. The right two columns show the output of our BEV-VE module and the ground truth flow in the BEV space.

This sample shows a driving scenario in a parking lot, in which most of the objects are static. Detection of CMT introduces large translation errors for predictions due to the ego vehicle motion. CMT+AsyncBEV corrects most of the static boxes and re-aligns the moving long bus. The color and saturation of the predicted flow align with the GT flow, which validates the interpretability of our proposed AsyncBEV.

## A.4 ABLATION OF BEV-VE DESIGN.

Table V ablates the effect of whether to use the multi-modal features (MM) and the flow loss (L) with the performance of all objects and dynamic objects under the $0.5\,s$ time offset. The basic setting in the first row also applies the concept of BEV-VE to explicitly predict the dynamic object flow and then warp the feature-coordinate mapping, but only takes the asynchronous features as input and is supervised by the detection loss in an end-to-end manner. The basic setting already achieves good perfor-

| Loss | MM | All Objects ($0.5s$) | | Dynamic Objects ($0.5s$) | |
|---|---|---|---|---|---|
| | | NDS | mAP | NDS | mAP |
| ✗ | ✗ | 63.0 | 56.8 | 37.0 | 25.4 |
| ✓ | ✗ | 63.2 ↑0.2 | 57.0 ↑0.2 | 37.6 ↑0.6 | 26.0 ↑0.6 |
| ✓ | ✓ | 63.3 ↑0.3 | 57.1 ↑0.3 | 37.8 ↑0.8 | 26.6 ↑1.2 |

Table V: Ablation of BEV-VE inputs. MM: Multimodal features, L: Flow loss.

mance. Using multi-modal features (MM) and the flow loss (Loss) for supervision further improves the performance under asynchrony with a small margin.

## A.5 ABLATION OF IMAGE BEV FEATURES.

Are the BEV features from SimpleBEV still necessary for a grid-based 3D object detector, which already builds BEV features from the multi-view image features? Table VI ablates the effect of the separate BEV encoder, which provides performance on All Objects and Dynamic Objects of UniBEV under LiDAR asynchrony with a $0.5\,s$ time offset. The first row shows the basic setting, in which our BEV-VE

| SimpleBEV | All Objects ($0.5\,s$) | | Dynamic Objects($0.5\,s$) | |
|---|---|---|---|---|
| | NDS | mAP | NDS | mAP |
| ✗ | 62.7 | 56.1 | 37.3 | 25.7 |
| ✓ | **63.3** ↑0.6 | **57.1** ↑1.0 | **37.8** ↑0.5 | **26.6** ↑0.9 |

Table VI: Ablation of SimpleBEV encoder

directly takes the BEV features for 3D object

detection as input, while the second row showcases that an extra SimpleBEV encoder is used specially for the $\Delta$-BEVFlow estimation. Despite the basic setting already achieving good performance with 62.7% NDS for All Objects and 37.3% NDS for Dynamic Objects, its counterpart with SimpleBEV consistently improves the performance with a small margin.

### A.6 Performance degradation for increasing time offsets

Figure VII further demonstrates the curves of mAP with increasing time offsets for three methods, vanilla CMT, CMT+EMC and CMT+AsyncBEV, which shares the same trend with Figure 4. Performance on all objects and dynamic objects is reported in the figure. CMT+AsyncBEV has the most robust performance against sensor asynchrony. Its performance in the extreme case with $0.5\ s$ is better than vanilla CMT with one-frame ($50\ ms$) asynchrony. Despite CMT+EMC delivering improved performance for All Objects, its performance for Dynamic Objects still shares the same degradation with vanilla CMT, which proves the necessity of a module to further compensate for the dynamic object motion.

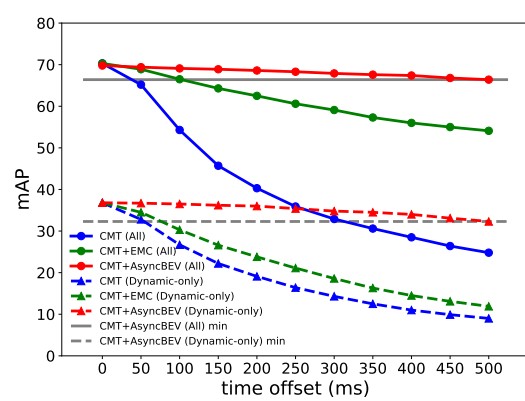

Figure VII: **Performance of CMT, CMT+EMC and CMT+AsyncBEV for different time offsets under LiDAR asynchrony.**

### A.7 Extension to multi-sensor asynchrony

The AsyncBEV framework can be extended to multi-sensor asynchrony scenarios. We presume to have one reference sensor (e.g., LiDAR) and treat all other (asynchronous) sensors (e.g., cameras and radars) as paired to the LiDAR reference only. This approach requires an additional lightweight AsyncBEV module for each additional asynchronous sensor modality, since network weights can be shared across sensors of the same modality. Possibly, multiple sensors of the same modality could be batched together (with a vector containing all their $\Delta t$) in a single forward pass to compute their $\Delta$-BEVFlow in parallel.

For the multi-sensor asynchrony scenario, the additional runtime and memory cost of AsyncBEV is negligible compared to the dominant cost from processing the extra sensor inputs. As shown in Table 1, AsyncBEV reduces the inference speed of CMT by only 0.4 FPS, and this overhead drops to 0.1 FPS when applied to a heavier baseline such as UniBEV. Furthermore, Table VII shows that the camera branch has $232\times$ the parameters of AsyncBEV, while the LiDAR branch has $23\times$. Therefore, when scaling to setups with more sensors, the backbone becomes substantially heavier, and the

| Module | # Params | $\times$AsyncBEV |
|---|---|---|
| Camera branch | $70, 572, 864$ | 232 |
| LiDAR branch | $6, 975, 184$ | 23 |
| AsyncBEV | $304, 114$ | 1 |

Table VII: # Params of CMT+AsyncBEV's modules. $\times$AsyncBEV is the parameter ratio of each module relative to AsyncBEV.

marginal runtime and memory overhead introduced by AsyncBEV becomes negligible compared to the total computation.

Finally, we note our experiments use all six cameras from nuScenes, treating them as a single sensor. Indeed, performance might futher improve if they are handled all independently, but our results already show the benefit of our approach even by grouping related asynchronous sensors.

## A.8 Dense GT flow Representation

The supervised scene flow task takes the ground truth flow as supervision to calculate a per-point flow for the source point cloud. The ground truth flow is usually generated by calculating the motion between bounding boxes (Jund et al., 2021; Zhang et al., 2024). Hence, this flow representation inherits the sparsity of point clouds. As point clouds usually only capture partial objects, this per-point sparse flow representation cannot fully present the object motion in a dense BEV grid. Therefore, we propose a dense ground truth flow representation, dense flow, which only relies on bounding boxes. By decoupling from point clouds, dense flow facilitates generating ground truth flow for camera BEV features.

The process to generate dense flow is shown in Algorithm 1. The core changes compared with the generation of scene flow ground truth are that we use a dense standard grid $\mathbb{G}_{\text{BEV}}$ to represent the dense and evenly distributed point cloud instead of using the real point cloud. Then we adopt a FlowEmbedder to voxelize $\mathbb{P}^{t_0}$ into the voxel space and scatter $\mathbb{V}_{\text{BEV}}^{t_0 \rightarrow t_1}$ with the voxelization indexes of $\mathbb{P}^{t_0}$ into the corresponding voxel cell, resulting in a dense flow representation $\mathbf{V}_{\text{BEV}}^{t_0 \rightarrow t_1}$, which spatially aligns with camera and LiDAR BEV features. $\mathbf{V}_{\text{BEV}}^{t_0 \rightarrow t_1}$ is used to supervise the predicted flow $\mathbf{V}_{flow}^{t_0 \rightarrow t_1}$.

## A.9 Algorithm to generate Dense Flow representation

Algorithm 1 summarizes the steps to generate a dense ground truth flow representation from bounding boxes, which serves as supervision for $\Delta$-BEVFlow.

---

**Algorithm 1** Generation of dense ground truth flow representation

---

**Require:** Bounding boxes at time $t_0$, $\mathbb{B}^{t_0}$; Bounding boxes at $t_1$, $\mathbb{B}^{t_1}$; ego motion $\mathbf{M}^{t_0 \rightarrow t_1}$; standard grid $\mathbb{G}_{BEV}$

1: **for** each box $b_i^{t_0} \sim \mathbb{B}^{t_0}$ **do**
2:      **for** each box $b_j^{t_1} \sim \mathbb{B}^{t_1}$ **do**          ▷ Boxes are defined in the global coordinate.
3:          **if** $b_i^{t_0}$.ID $==$ $b_j^{t_1}$.ID **then**
4:              $T_{box2g}^{t_1} \leftarrow b_j^{t_1}$.center, $b_j^{t_1}$.orientation
5:              $T_{box2g}^{t_0} \leftarrow b_i^{t_0}$.center, $b_i^{t_0}$.orientation
6:              $T_{l2e}^{t_0}, T_{e2g}^{t_0} \leftarrow \mathbf{M}^{t_0 \rightarrow t_1}$
7:              $T_{box2t_0}^{t_1} \leftarrow (T_{l2e}^{t_0})^{-1} \cdot (T_{e2g}^{t_0})^{-1} \cdot T_{box2g}^{t_1}$   ▷ Transfer the box at $t_1$ to LiDAR frame at $t_0$
8:              $T_{box2t_0}^{t_0} \leftarrow (T_{l2e}^{t_0})^{-1} \cdot (T_{e2g}^{t_0})^{-1} \cdot T_{box2g}^{t_0}$   ▷ Transfer the box at $t_0$ to LiDAR frame at $t_0$
9:              $R_{t_0 2t_1} \leftarrow T_{box2t_0}^{t_1} \cdot (T_{box2t_0}^{t_0})^{-1}$      ▷ Calculate the motion of boxes from $t_0$ to $t_1$
10:              $\mathbb{P}_{b_i^{t_0}} \leftarrow f_{pts\_in\_boxes}(b_i^{t_0}, \mathbb{G}_{BEV})$          ▷ Get grid points in the box at $t_0$
11:              $\mathbb{V}_{b_i^{t_0}}^{t_0 \rightarrow t_1} \leftarrow R_{t_0 2t_1} \cdot \mathbb{P}_{b_i^{t_0}} - \mathbb{P}_{b_i^{t_0}}$   ▷ Calculate the scene flow for the points within box
12:
13:          **end if**
14:      **end for**
15: **end for**
16: **return** $\mathbb{V}_{BEV}^{t_0 \rightarrow t_1} \leftarrow \{\mathbb{V}_{b_i^{t_0}}^{t_0 \rightarrow t_1}, b_i^{t_0} \in \mathbb{B}^{t_0}\}$, $\mathbb{P}^{t_0} \leftarrow \{\mathbb{P}_{b_i^{t_0}}, b_i^{t_0} \in \mathbb{B}^{t_0}\}$

---

## A.10 The Use of Large Language Models (LLMs)

Large Language Models (LLMs) are used only to polish the grammar and phrasing of sentences and to search for potentially relevant prior works. All conceptual development, experimental design, implementation, analysis, and initial writing of the paper were carried out by the authors.

