# OpenReview forum: "AsyncBEV: Cross-modal flow alignment in Asynchronous 3D Object Detection"
_ICLR.cc/2026/Conference — ICLR 2026 Poster_

### Official Review · Reviewer_gWRm · 2025-10-27

**Soundness:** 3
**Presentation:** 3
**Contribution:** 3
**Rating:** 6
**Confidence:** 4

**Summary:**

This paper proposes AsyncBEV, a lightweight module designed to enhance the robustness of multi-modal 3D object detectors against sensor asynchrony by introducing a novel Δ-BEVFlow estimation task. The method predicts motion in BEV feature space and warps asynchronous features to a reference timestamp, demonstrating compatibility with both token-based and grid-based detectors. The writing is clear and well-structured, and the experimental evaluation on nuScenes shows significant improvements in handling asynchronous inputs, especially for dynamic objects.

**Strengths:**

1.The paper is clearly written and easy to follow.

2.It explores an important and underexplored problem in multi-modal perception.

3.The method is simple yet effective, with a lightweight and generic design.

4.Extensive experiments validate the effectiveness of the proposed approach under various asynchronous settings.

**Weaknesses:**

1.The validation is limited to the nuScenes dataset, lacking cross-dataset generalization.

2.The method does not address scenarios where multiple sensors are asynchronous at the same time, which may lead to increased computational complexity.

3.The Motion Compensation-based baseline is somewhat weak. The comparison to asynchronous fusion methods in related work, like StreamingFlow and TimeAlign, is missing.

**Questions:**

1.Have the authors considered evaluating AsyncBEV on other autonomous driving datasets, such as Waymo, to further demonstrate its generalization capability?

2.How would the method scale and perform when more than two sensors are asynchronous simultaneously, and would the current flow estimation strategy still be effective?

---

> ### Author Response · Authors · 2025-11-21
> **[W1&Q1] Generalization across datasets**
>
> Thank you for your valuable observations and questions, we very much appreciate your review.
>
> ---
>
> This is a valid suggestion, and we did indeed consider validating our method on another dataset before submitting our paper. Please let us explain our considerations for focusing on nuScenes only.
>
> We intended to explore other datasets with our strongest baseline, CMT, which reports results for nuScenes and Argoverse. Unfortunately, no pre-trained models were available for Argoverse, and reproducing the CMT results on that dataset would require significant additional effort to produce a properly trained Argoverse baseline (we found this is a common problem, as others have also asked for a CMT Argoverse configuration on the official Github, but no one has been able to provide one [1]). Furthermore, our UniBEV baseline only provides pretrained results on nuScenes too. Since our experimental design aims to validate our approach on these diverse baseline networks, we have opted to focus our in-depth experiments and available compute on nuScenes only at this stage.
>
> [1] https://github.com/junjie18/CMT/issues/91

---

> ### Author Response · Authors · 2025-11-21
> **[W2&Q2] Multi-sensor Asynchrony**
>
> Yes, we expect our flow prediction strategy should still work.
>
> We should clarify the mentioned intended future work: We presume to have one reference sensor (e.g., Lidar) and treat all other (asynchronous) sensors (e.g., cameras and radars) as paired to the lidar reference only. This approach requires one more lightweight AsyncBEV module for each additional asynchronous sensor modality, since network weights can be shared between sensors of the same modality. Possibly, multiple sensors of the same modality could be batched (with a vector containing all their Δt) in a single forward pass to compute their Δ-BEVFlow in parallel. For the multi-sensor asynchrony scenario, the additional runtime and memory cost of AsyncBEV is negligible compared to the dominant cost from processing the extra sensor inputs. As shown in Table 1, AsyncBEV reduces the inference speed of CMT by only 0.4 FPS, and this overhead drops to 0.1 FPS when applied to a heavier baseline such as UniBEV. Furthermore, Table A shows that the camera branch has **232×** the parameters of AsyncBEV, while the LiDAR branch has **23×**. Therefore, when scaling to setups with more sensors, the backbone becomes substantially heavier, and the marginal runtime and memory overhead introduced by AsyncBEV becomes negligible compared to the total computation. We shall add this to the paper’s Appendix.
>
> *Table A: # Parameters of CMT+AsyncBEV*
>
> | Module | # Params|
> |----------|----------:|
> | camera branch | 70,572,864|
> | LiDAR branch | 6,975,184|
> | AsyncBEV|304,114 |
>
> Finally, we note our experiments also use all 6 cameras from nuScenes, treating them as a single sensor. Indeed, performance might even improve further if they are handled all independently, but our results already show the benefit of our approach even by grouping related asynchronous sensors.

---

> ### Author Response · Authors · 2025-11-21
> **[W3] Missing comparison to other asynchronous fusion methods**
>
> Thanks for pointing it out. StreamingFlow [1] and TimeAlign [2] share a similar concept, i.e., using an RNN-based network to model the temporal dynamics from several history frames and update the asynchronous features to the reference timestamp. As TimeAlign does not provide any code, we focused our effort on reproducing StreamingFlow as an additional experimental baseline.
>
> The core concept of StreamingFlow is utilizing a GRU-ODE to model the temporal dynamics among grid-like features from different timestamps. To adapt StreamingFlow to object detection, we can only integrate the GRU-ODE module into our grid-based baseline, UniBEV. Table C compares UniBEV+StreamingFlow to UniBEV+AsyncBEV under LiDAR asynchrony following the regular experimental setup, i.e. one pair of asynchronous multi-modal observations as input. As shown in the table, the UniBEV+StreamingFlow approach performs worse than UniBEV+AsyncBEV. We shall include this baseline to our paper, and add it to Table 1.
>
> *Table C. Performance of StreamingFlow on All Objects when LiDAR is asynchronous*
> | Method   |  NDS (0s) | NDS (0.25s) |NDS (0.5s) |FPS|
> |----------|:----:|:----:|:----:|:----:|
> | UniBEV+StreamingFlow  | 62.8 | 55.8 | 50.0| 1.9|
> | UniBEV+AsyncBEV| 65.7 | 64.8 | 63.3| 2.7|
>
> ---
>
> [1] Shi, Yining, et al. "Streamingflow: Streaming occupancy forecasting with asynchronous multi-modal data streams via neural ordinary differential equation." In *CVPR 2024*. \
> [2] Song, Zhihang, et al. "Timealign: A multi-modal object detection method for time misalignment fusing in autonomous driving." *arXiv preprint arXiv:2412.10033 (2024)*.

---

> ### Author Response · Authors · 2025-12-03
> **[W3] Further comparison with multi-frame StreamingFlow**
>
> In the previous post, we reported the results of StreamingFlow, which uses the same input data format as AsyncBEV for both training and validation, namely one pair of asynchronous multi-modal observations (i.e., two observations at different timestamps). This variant of StreamingFlow is referred to as StreamingFlow* in future discussions.
>
> Since StreamingFlow’s GRU-ODE module benefits from long sequences of input data to model the temporal dynamics of BEV features, we now also tested a different experimental setup of StreamingFlow†, which takes as input additional historic frames. Following the setup in SteamingFlow’s codebase, we incorporate two additional history LiDAR inputs and two additional history image inputs to construct a sequence of history observations at four different prior timestamps. With the default two asynchronous observations, StreamingFlow† takes a sequence of 6 observations as input. We added the results of StreamingFlow† now to Table D, see below.
>
> *Table D. Performance of StreamingFlow on All Objects when LiDAR is asynchronous.*
>
> | Method   | Train input| Infer input| NDS (0s) | NDS (0.25s) |NDS (0.5s) |FPS|
> |----------|:----:|:----:|:----:|:----:|:----:|:----:|
> | UniBEV+StreamingFlow*  | async input| async input| 62.8 | 55.8 | 50.0| 1.9|
> | UniBEV+StreamingFlow†  | async input + history data |async input + history data |64.2 | 58.8 | 54.1| 1.0|
> | UniBEV+AsyncBEV		  | async input| async input|65.7 | 64.8 | 63.3| 2.7|
>
> In Table D, 'async input' denotes a single pair of asynchronous multi-modal observations, while the 'history data' represents the extended history of multi-modal observations across four timestamps.
>
> Through the addition of historic data StreamingFlow† improves performance for both synchronous and asynchronous scenarios compared to StreamingFlow*, as it improves motion prediction. Nevertheless, in both scenarios StreamingFlow† still underperforms our AsyncBEV which only uses the latest asynchronous observations as input. Furthermore, due to the need to process every observation in the input sequence with the GRU-ODE module, the FPS of StreamingFlow† is much lower than AsyncBEV. Overall, Table D highlights the superior efficiency and effectiveness of our AsyncBEV compared with StreamingFlow.

---

### Official Review · Reviewer_zA9G · 2025-10-31

**Soundness:** 3
**Presentation:** 3
**Contribution:** 2
**Rating:** 8
**Confidence:** 4

**Summary:**

This work presents a novel framework for improving the robustness of 3D object detection models in the presence of asynchronous sensors. The proposed framework consists of AsyncBEV, a lightweight and generic module that can be integrated into existing BEV object detection models for improved robustness against sensor asynchrony. AsyncBEV estimates the 2D flow in the BEV feature space and then warps the asynchronous sensor data to align with a reference frame. Extensive experiments on the nuScenes dataset with both grid-based and token-based BEV detectors show the effectiveness of the proposed approach in varying levels of asynchrony between LiDAR and camera sensors, especially for dynamic objects.

**Strengths:**

- Addressing robustness against sensor asynchrony is important from a practical viewpoint since autonomous vehicles often come equipped with multiple sensors.
- The paper is well-written and easy to follow. The description is detailed and the figures (Fig.2,3,5) are informative.
- AsyncBEV module is a generic and lightweight module that can be combined with both grid-based and token-based BEV detectors.
- The proposed module predicts the delta 2D scene flow in BEV space, conditioned on delta timesteps across multimodal BEV features.
- Experiments on nuScenes (Tab.1) show the benefits of AsyncBEV over egomotion compensation (EMC) on both token-based (CMT) and grid-based (UniBEV) approaches.
- Ablations in Fig.4, Tab.2 provide more insights into the capabilities of different components.

**Weaknesses:**

- In Sec.4.2, for the finetuning UniBEV variant, is the delta timestep offset (between reference and asynchronous sensor) also used as input? It'd be useful to have a finetuning baseline that also incorporates delta timestep offset. For example, LiDAR BEV can be augmented with the delta timestep (on a per-point basis) as an additional feature channel (a similar strategy was also used in the Fan et al. 2025 referenced paper). This would help understand if the delta flow formulation is indeed effective compared to simpler alternatives like finetuning with the delta timestep as an additional feature channel.
- The finetuning variant in Tab.2 should also be a baseline in Tab.1 (applied to both CMT and UniBEV) since it leads to extensive gains (as noted in Tab.2). This is relevant since CMT and UniBEV are not trained with any asynchronous data.
- Since EMC is quite widely used in the autonomous driving literature (as mentioned in the paper), it'd be useful to report the delta gains with respect to EMC variants in Tab.1. The gains of the proposed approach are still clear, this would better contextualize the benefits of AsyncBEV over the standard EMC approach.

**Questions:**

The paper is well written, and the claims are validated in the experiments. My main concern is regarding simpler alternatives as baselines to better contextualize the benefits of the delta flow formulation (more details in the weaknesses above):
- A finetuning variant for both CMT & UniBEV should be added to Tab.1 since these methods are not trained on asynchronous data.
- A finetuning variant where the delta timestep offset is used as additional input should also be considered.

---

> ### Author Response · Authors · 2025-11-21
> **[W1&Q2] Comparison with a finetuning variant using Δt features**
>
> Thank you for your support for our work and for your detailed and relevant comments. We appreciate your effort, and address your remaining questions in our responses below.
>
> ---
>
> Similar to the work of Fan et al [1], we concatenate the asynchronous time offset to the input point cloud. Table A demonstrates its performance on All Objects when LiDAR is asynchronous. Despite CMT+Fan et al performing worse in the synchronous scenarios, it outperforms CMT+EMC under sensor asynchrony. However, the overall performance is still inferior to that of our proposed AsyncBEV. We will include this table in our paper.
>
> *Table A. Performance of Fan et al. on All Objects when LiDAR is asynchronous*
> | Method   |  NDS (0s) | NDS (0.25s) |NDS (0.5s) |
> |----------|:----:|:----:|:----:|
> | CMT+EMC  | 72.9 | 66.8 | 63.3|
> | CMT+Fan et al.|70.6|70.0|66.2|
> | CMT+AsyncBEV| 72.5 | 71.5 | 70.0|
>
> ---
>
> [1] Fan, Meng, et al. "Robust sensor fusion against on-vehicle sensor staleness." In *CVPRW 2025*.

---

> ### Author Response · Authors · 2025-11-21
> **[W2&W3&Q1] Add more baselines to Table 1**
>
> Thank you very much for these suggestions, we agree and will adapt the paper accordingly. We will add the experiments for vanilla model finetuning with asynchronous data into Table 1. Meanwhile, we will adjust Table 1 to show the delta gains over EMC. The PDF will be updated by the December 3 deadline.

---

### Official Review · Reviewer_LiTB · 2025-11-01

**Soundness:** 3
**Presentation:** 3
**Contribution:** 3
**Rating:** 6
**Confidence:** 4

**Summary:**

This paper addresses a critical challenge in autonomous driving perception: sensor asynchrony (caused by mismatched sensor frequencies, network latency, or hardware bottlenecks). Most 3D object detectors rely on perfectly synchronized multi-modal data (LiDAR/camera), but asynchrony degrades performance—especially for dynamic objects, which are key to safety. To solve this, the authors propose AsyncBEV, a lightweight, generic module compatible with both token-based (e.g., CMT) and grid-based (e.g., UniBEV) detectors.
AsyncBEV introduces Δ-BEVFlow estimation, a novel task that predicts 2D flow in Bird’s Eye View (BEV) space between multi-modal features using known time offsets. Unlike Ego Motion Compensation (EMC, which only aligns static objects) or traditional scene flow methods (requiring point clouds and fixed offsets), Δ-BEVFlow explicitly models dynamic object motion. It offers two formulations: motion-based (direct flow regression) and velocity-based (predict velocity first, then scale by time)—the latter is adopted for better regularization.
Experiments on the nuScenes dataset show AsyncBEV significantly enhances robustness: in the worst-case 0.5s time offset, it outperforms EMC baselines by 16.6% (CMT) and 11.9% (UniBEV) in NDS for dynamic objects, with minimal computational overhead (marginal FPS loss).

**Strengths:**

**High Practical Relevance**

Sensor asynchrony is unavoidable in real-world autonomous driving, yet it is often overlooked in detector design. By targeting this gap, AsyncBEV directly improves the safety and reliability of perception systems—particularly for dynamic objects (e.g., pedestrians, moving vehicles), which are the primary cause of accidents. This makes the work valuable for both academic research and industrial deployment.

**Effective Δ-BEVFlow Design**

Δ-BEVFlow addresses key limitations of prior methods: it operates on multi-modal BEV features (not raw point clouds) and supports variable time offsets, enabling flexible cross-modal alignment. The velocity-based formulation further strengthens the design: by decoupling velocity from time, it ensures flow approaches zero for near-synchronous sensors (avoiding unnecessary distortions) and simplifies learning, as validated by ablation studies (Table 2).

**Generality and Lightweight Integration**

AsyncBEV is architecture-agnostic: it adapts to token-based detectors (by adjusting token coordinates) and grid-based detectors (by generating grid look-up tables) with minimal modifications. It also introduces negligible computational overhead—experiments show only a 0.3–0.4 FPS drop (Table 1)—making it suitable for real-time autonomous driving pipelines.

**Strong Interpretability**

Explicit flow prediction allows intuitive visualization (Figures 5, 6), where predicted Δ-BEVFlow closely aligns with ground truth and directly corrects bounding box misalignments. Quantitative results (e.g., Figure 4’s flat performance curves for AsyncBEV) further confirm its robustness across varying time offsets, enhancing trust in the module’s mechanism.

**Weaknesses:**

**Limited Novelty Compared to Prior Asynchrony-Robust Work**

The core idea of using BEV flow for asynchrony compensation is not entirely new. For example, CoBEVFlow (Wei et al., NeurIPS 2023) already uses BEV flow to handle asynchronous collaborative perception, though it relies on object proposals and is more computationally heavy. Additionally, recent work like UniV2X (Yu et al., AAAI 2025) explores end-to-end autonomous driving with V2X cooperation, which also involves addressing multi-agent asynchrony. The paper acknowledges these works but does not sufficiently emphasize how Δ-BEVFlow advances beyond them—e.g., why feature-based flow (without proposals) offers better generalization, or how variable time offset handling outperforms alternatives. This weakens the claim of novelty.

**Simplified Asynchrony Assumptions**

The paper assumes only two sensors (one synchronous reference, one asynchronous), but real autonomous driving systems use 5–10 sensors (e.g., 6 cameras, 1 LiDAR, 5 radars in nuScenes). AsyncBEV cannot handle scenarios where 3+ sensors have overlapping offsets (e.g., LiDAR delayed by 0.2s, front camera by 0.1s). The authors mention extending to multi-sensor asynchrony as future work, but the current design lacks scalability to full-scale vehicle perception pipelines.

**Synchronous Performance Trade-Off**

Default AsyncBEV causes small but consistent performance drops in the synchronized case (0s offset): 0.4% NDS for CMT and 1.0% NDS for UniBEV (Table 1). The "frozen detector" variant (AsyncBEV-FD) avoids this drop but underperforms in asynchronous scenarios. The paper does not explore alternative training strategies (e.g., adaptive loss weighting) to resolve this trade-off—critical for deployment, as synchronized sensors are the most common real-world scenario.

**Questions:**

**Limited Novelty Compared to Prior Asynchrony-Robust Work**

The core idea of using BEV flow for asynchrony compensation is not entirely new. For example, CoBEVFlow (Wei et al., NeurIPS 2023) already uses BEV flow to handle asynchronous collaborative perception, though it relies on object proposals and is more computationally heavy. Additionally, recent work like UniV2X (Yu et al., AAAI 2025) explores end-to-end autonomous driving with V2X cooperation, which also involves addressing multi-agent asynchrony. The paper acknowledges these works but does not sufficiently emphasize how Δ-BEVFlow advances beyond them—e.g., why feature-based flow (without proposals) offers better generalization, or how variable time offset handling outperforms alternatives. This weakens the claim of novelty.

**Simplified Asynchrony Assumptions**

The paper assumes only two sensors (one synchronous reference, one asynchronous), but real autonomous driving systems use 5–10 sensors (e.g., 6 cameras, 1 LiDAR, 5 radars in nuScenes). AsyncBEV cannot handle scenarios where 3+ sensors have overlapping offsets (e.g., LiDAR delayed by 0.2s, front camera by 0.1s). The authors mention extending to multi-sensor asynchrony as future work, but the current design lacks scalability to full-scale vehicle perception pipelines.

**Synchronous Performance Trade-Off**

Default AsyncBEV causes small but consistent performance drops in the synchronized case (0s offset): 0.4% NDS for CMT and 1.0% NDS for UniBEV (Table 1). The "frozen detector" variant (AsyncBEV-FD) avoids this drop but underperforms in asynchronous scenarios. The paper does not explore alternative training strategies (e.g., adaptive loss weighting) to resolve this trade-off—critical for deployment, as synchronized sensors are the most common real-world scenario.

---

> ### Author Response · Authors · 2025-11-21
> **[W1&Q1] Limited Novelty Compared to Prior Asynchrony-Robust Work**
>
> Thank you for your time and effort. We appreciate your assessment and insightful remarks. We will respond to your questions below.
>
> ---
>
> Thank you for pointing out UniV2X [2]. We note that CoBEVFlow [1] and UniV2X [2] are designed for collaborative perception on distributed but homogeneous sensor networks. CoBEVFlow first generates object proposals from distinct agent features and then applies an attention-based module to associate the proposals from different agents. It subsequently calculates flows based on the associated object proposals. Hence, the flow in CoBEVFlow is not directly learned. UniV2X applies an agent query-based flow predictor, taking the associated agent from multiple frames as input.
>
> In contrast, AsyncBEV has several substantial differences to these two methods:
> - CoBEVFlow and UniV2X calculate flow based on the object proposals, therefore their performance heavily relies on the quality of the generated object proposals. However, in non-homogeneous 3D object detection, the detection characteristics for different sensors can differ drastically. For instance, camera detectors usually perform worse than LiDAR detectors. The inconsistent prediction of object proposals across modalities would affect subsequent feature alignment, object association, and flow calculation. In contrast, our Δ-BEVFlow is estimated directly from the multi-modal features, avoiding the chicken-and-egg problem of detecting objects before aligning features to detect objects.
> - Proposal-based flow propagates and accumulates errors from the proposal prediction. For example, the FP and FN proposals increase the computational burden and complexity of object association in CoBEVFlow. Furthermore, proposal-based flow only learns information from a single potential object. Our feature-based flow can learn richer information from all multi-modal features, including environmental context.
> - Even assuming correct object proposals, in these two works all agents send homogeneous features from the same modality (LiDAR features in CoBEVFlow [1] and image features in UniV2X [2]). In contrast, AsyncBEV is designed to align heterogeneous multi-modal features. The distinct features pose further challenges in associating object proposals generated from different sensors [1] or predicting query flows with a single MLP module [2]. The effectiveness of their temporal alignment method has not been validated in a multi-modal setup.
>
> We will include these works in the related work, and emphasize in the paper the highlighted novelty and value of Δ-BEVFlow to both these prior works.
>
> ---
>
> [1] Wei, Sizhe, et al. "Asynchrony-robust collaborative perception via bird's eye view flow." in *NeurIPS 2023*.\
> [2] Yu, Haibao, et al. "End-to-end autonomous driving through v2x cooperation." in *AAAI  2025*.

---

> ### Author Response · Authors · 2025-11-21
> **[W2&Q2] Simplified Asynchrony Assumptions**
>
> We should clarify the mentioned intended future work: We presume to have one reference sensor (e.g., Lidar) and treat all other (asynchronous) sensors (e.g., cameras and radars) as paired to the lidar reference only. This approach requires one more lightweight AsyncBEV module for each additional asynchronous sensor modality, since network weights can be shared between sensors of the same modality. Possibly, multiple sensors of the same modality could be batched (with a vector containing all their Δt) in a single forward pass to compute their Δ-BEVFlow in parallel. For the multi-sensor asynchrony scenario, the additional runtime and memory cost of AsyncBEV is negligible compared to the dominant cost from processing the extra sensor inputs. As shown in Table 1, AsyncBEV reduces the inference speed of CMT by only 0.4 FPS, and this overhead drops to 0.1 FPS when applied to a heavier baseline such as UniBEV. Furthermore, Table A shows that the camera branch has **232×** the parameters of AsyncBEV, while the LiDAR branch has **23×**. Therefore, when scaling to setups with more sensors, the backbone becomes substantially heavier, and the marginal runtime and memory overhead introduced by AsyncBEV becomes negligible compared to the total computation. We shall add this to the paper’s Appendix.
>
> *Table A: \# Parameters of CMT+AsyncBEV*
>
> | Module | \# Params|
> |----------|----------:|
> | camera branch | 70,572,864|
> | LiDAR branch | 6,975,184|
> | AsyncBEV|304,114 |
>
> Finally, we note our experiments also use all 6 cameras from nuScenes, treating them as a single sensor. Indeed, performance might even improve further if they are handled all independently, but our results already show the benefit of our approach even by grouping related asynchronous sensors.

---

> ### Author Response · Authors · 2025-11-21
> **[W3&Q3] Synchronous Performance Trade-off**
>
> Indeed, there is a small trade-off. We do note that 0.4% is small compared to the larger performance differences introduced by fundamentally different system architectures such as CMT and UniBEV.
> Leveraging the adaptive loss weighting is a good suggestion, and a promising direction for future work. Unfortunately, we have not been able to pursue this direction during this rebuttal period due to time constraints.
>
> We expect that trade-off could also be improved in favor of the more common 0s offset (synchronous) case by modifying the Δt distribution of samples in the training dataset, e.g. by mixing 50% synchronous (0s) and 50% asynchronous (> 0s) scenarios, or some other weights. We opted to use a uniform distribution in our current experiments for its simplicity: It avoids introducing additional experimental parameters for this distribution, which one could also tune, ablate, etc., as that would further complicate discussion of the results.

---

### Official Review · Reviewer_VrJt · 2025-11-01

**Soundness:** 3
**Presentation:** 3
**Contribution:** 2
**Rating:** 4
**Confidence:** 4

**Summary:**

This paper introduces AsyncBEV, a module designed to improve the robustness of LiDAR–camera 3D object detection under asynchrony. The proposed module predicts a Δ-BEVFlow, a BEV flow to spatially align asynchronous lidar features prior to fusion. The approach generalizes across both grid and token-based BEV detectors and improves performance over ego-motion compensation on nuScenes, particularly for dynamic objects and large temporal offsets (0.1–0.5 s). While technically sound and empirically validated, the work would benefit from stronger comparison against alternative approaches to help motivate the usefulness.

**Strengths:**

- Conceptually intuitive and interpretable formulation for feature alignment under temporal misalignment and dynamic motion.
- Generalizable design compatible with both grid and token BEV frameworks.
- Consistent improvements across offset magnitudes with negligible runtime overhead.

**Weaknesses:**

- Lacks discussion of real-world latency handling in AV stacks, where stale data (>100 ms) are typically discarded. Comparison to such baselines (e.g., camera-only inference or temporal propagation) would clarify practical value.
- Limited benchmarking against contemporary temporal alignment methods like StreamingFlow.
- Performance could be impacted by stochastic latency profiles, but no analysis is provided on sensitivity to time offset estimation error.

**Questions:**

- How does AsyncBEV compare to fallback strategies such as camera-only inference or simple temporal propagation when LiDAR is delayed or missing?
- What are the performance and latency trade-offs relative to recent streaming or time-aware fusion methods such as StreamingFlow?
- Could Δ-BEVFlow be trained to handle uncertain or estimated time offsets instead of relying on ground-truth Δt?

---

> ### Author Response · Authors · 2025-11-21
> **[W1&Q1] Comparison with fallback strategy and temporal propagation**
>
> Thank you for your insightful review and valuable suggestions. You raise good points, which we hope will be addressed by the additional results and analysis below.
>
> ---
>
> **Sensor fallback strategy**
>
> In our 0.25s and 0.5s delay setups, discarding stale data (>100 ms) is equivalent to single-modality inference. Table A below compares our AsyncBEV to single-modality on All Objects. This comparison, including the mAP metric and the Static/Dynamic categories, will be added to the paper before the December 3 deadline.
>
> *Table A. Single-modality performance of CMT and UniBEV*
> | Method   |  NDS |
> |----------|:----:|
> | CMT_C    | 44.7 |
> | CMT+AsyncBEV (async L 0.5s) | 70.0 (↑25.3) |
> | UniBEV_C | 42.6 |
> | UniBEV+AsyncBEV(Async L 0.5s) | 63.3 (↑20.7)|
> | CMT_L    |68.1  |
> | CMT+AsyncBEV (Async C 0.5s) | 72.3 (↑4.2)|
> | UniBEV_L | 63.7 |
> | UniBEV+AsyncBEV(Async C 0.5s)| 66.7 (↑3.0)|
>
> As shown in Table A, when lidar is delayed and a sensor fallback strategy is adopted, camera-only CMT_C achieves 44.7% NDS, while camera-only UniBEV_C performs at 42.6% NDS. However, our CMT+AsyncBEV achieves 70.0% NDS, and UniBEV achieves 63.3% NDS even with an extreme 0.5s time offset for LiDAR, outperforming the single-modality performance significantly.
>
> We make similar observations for the camera-delayed scenario. With a 0.5s time offset for cameras, CMT+AsyncBEV achieves 72.3% NDS, while UniBEV+AsyncBEV achieves 66.7% NDS, both surpassing the LiDAR-only performance of CMT_L (68.1% NDS) and UniBEV_L (63.7% NDS). This shows that the asynchronous modality still contains rich information for perception tasks. In conclusion, effectively leveraging temporally misaligned information is a better option than completely dropping them.
>
> ---
>
> **Temporal propagation strategy**
>
> Our Ego Motion Compensation (EMC) baseline can be regarded as a simple temporal propagation method. With known ego motions from two timestamps, the asynchronous features are propagated into the reference coordinate system. The improved performance of AsyncBEV over EMC has already been discussed in the paper.
>
>
> The approach proposed by Fan et al. [1] can be considered an alternative temporal propagation strategy. Their approach simply concatenates the async time offsets Δt to each point in the point cloud provided as input to the network. The network is expected to learn to align the features implicitly according to the given asynchronous time offset conditions. We have now implemented this concept from [1] on our baselines, and report the performance in Table B. We will include the results in our paper by December 3rd. The temporal propagation strategy proposed by Fan et al. shows a significant drop in performance for the 0s case, but does mitigate some of the misalignment for higher time offsets compared to the EMC baseline. However, its performance is still inferior to our proposed AsyncBEV for all time offsets, with the performance gap increasing as the time offset increases.
>
> *Table B. Performance of time offsets propagation on All Objects when LiDAR is asynchronous*
> | Method   |  NDS (0s) | NDS (0.25s) |NDS (0.5s) |
> |----------|:----:|:----:|:----:|
> | CMT+EMC  | 72.9 | 66.8 | 63.3|
> |CMT+Fan et al.|70.6|70.0|66.2|
> | CMT+AsyncBEV (ours)| 72.5 | 71.5 | 70.0|
>
> ---
>
> [1] Fan, Meng, et al. "Robust sensor fusion against on-vehicle sensor staleness." In *CVPRW 2025*.

---

> ### Author Response · Authors · 2025-11-21
> **[W2&Q2] Comparison with StreamingFlow**
>
> Thank you for pointing out StreamingFlow [2]. StreamingFlow is presented and evaluated to address an occupancy forecasting task for future time steps.
>
> Indeed, StreamingFlow shares a similar motivation to our method for temporal alignment in feature space. Nevertheless, it has various key differences from our approach:
> + As StreamingFlow is designed for forecasting, it takes a sequence of past images or LiDAR frames as input, and predicts features for a (future) reference frame without the reference observation and then they fuse the reference frame features *after* this motion prediction. In contrast, AsyncBEV utilizes paired multi-modal observations from both past *and* target reference timestamps. Our experiments in Table IV (Appendix) confirm using the reference observation to align features is beneficial.
> + StreamingFlow does not produce an explicit flow field, whereas AsyncBEV does through its Δ-BEVFlow. Such explicit flow fields can be valuable for other autonomous driving tasks too, as witnessed by the diverse work on scene flow estimation.
> + StreamingFlow takes streaming multi-modal features as a set of observations, which requires a unified feature representation, such as grid-like BEV features, for both modalities. Fusion is restricted to the update process of StreamingFlow GRU-ODE. In contrast, AsyncBEV can be applied to both token-based and grid-based BEV methods, due to its explicit flow. It is a plug-and-play module which can be integrated in various existing fusion network designs, which is more flexible. We show this by integrating AsyncBEV in both CMT and UniBEV.
>
> We now test these differences experimentally. The core concept of StreamingFlow is utilizing a GRU-ODE to model the temporal dynamics among grid-like features from different timestamps. To adapt StreamingFlow to object detection, we can only integrate the GRU-ODE module into our grid-based baseline, UniBEV. Table C compares UniBEV+StreamingFlow to UniBEV+AsyncBEV under LiDAR asynchrony following the regular experimental setup, i.e. one pair of asynchronous multi-modal observations as input. As shown in the table, the UniBEV+StreamingFlow approach performs worse than UniBEV+AsyncBEV. We shall include this baseline to our paper, and add it to Table 1.
>
> *Table C. Performance of StreamingFlow on All Objects when LiDAR is asynchronous*
> | Method   |  NDS (0s) | NDS (0.25s) |NDS (0.5s) |FPS|
> |----------|:----:|:----:|:----:|:----:|
> | UniBEV+StreamingFlow  | 62.8 | 55.8 | 50.0| 1.9|
> | UniBEV+AsyncBEV| 65.7 | 64.8 | 63.3| 2.7|
>
> ---
>
> [2] Shi, Yining, et al. "Streamingflow: Streaming occupancy forecasting with asynchronous multi-modal data streams via neural ordinary differential equation." In *CVPR 2024*.

---

> ### Author Response · Authors · 2025-11-21
> **[W2&Q2] Comparison with multi-frame StreamingFlow**
>
> In the previous post, we reported the results of StreamingFlow, which uses the same input data format as AsyncBEV for both training and validation, namely one pair of asynchronous multi-modal observations (i.e., two observations at different timestamps). This variant of StreamingFlow is referred to as StreamingFlow* in future discussions.
>
> Since StreamingFlow’s GRU-ODE module benefits from long sequences of input data to model the temporal dynamics of BEV features, we now also tested a different experimental setup of StreamingFlow†, which takes as input additional historic frames. Following the setup in SteamingFlow’s codebase, we incorporate two additional history LiDAR inputs and two additional history image inputs to construct a sequence of history observations at four different prior timestamps. With the default two asynchronous observations, StreamingFlow† takes a sequence of 6 observations as input. We added the results of StreamingFlow† now to Table D, see below.
>
> Table D. Performance of StreamingFlow on All Objects when LiDAR is asynchronous.
> | Method   | Train input| Infer input| NDS (0s) | NDS (0.25s) |NDS (0.5s) |FPS|
> |----------|:----:|:----:|:----:|:----:|:----:|:----:|
> | UniBEV+StreamingFlow*  | async input| async input| 62.8 | 55.8 | 50.0| 1.9|
> | UniBEV+StreamingFlow†  | async input + history data |async input + history data |64.2 | 58.8 | 54.1| 1.0|
> | UniBEV+AsyncBEV		  | async input| async input|65.7 | 64.8 | 63.3| 2.7|
>
> In Table D, 'async input' denotes a single pair of asynchronous multi-modal observations, while the 'history data' represents the extended history of multi-modal observations across four timestamps.
>
> Through the addition of historic data StreamingFlow† improves performance for both synchronous and asynchronous scenarios compared to StreamingFlow*, as it improves motion prediction. Nevertheless, in both scenarios StreamingFlow† still underperforms our AsyncBEV which only uses the latest asynchronous observations as input. Furthermore, due to the need to process every observation in the input sequence with the GRU-ODE module, the FPS of StreamingFlow† is much lower than AsyncBEV. Overall, Table D highlights the superior efficiency and effectiveness of our AsyncBEV compared with StreamingFlow.

---

> ### Author Response · Authors · 2025-11-21
> **[W3&Q3] Performance with uncertain Δt**
>
> This is an interesting point. We expect that the timestamp information reported by individual sensors is reasonably accurate for the moments they capture in an autonomous driving stack, as sensor hardware is typically responsible for providing a correct timestamp with each measurement, using established clock alignment protocols. Additionally, we note that all our experiments on nuScenes have used real-world sensor timestamps for the Δt calculation. Thus, our results already contain real-world misalignment between the timestamps and sensor data itself.
>
>
> Still, as a new experiment, we now tried evaluating our model on Δt with added Gaussian noise, simulating time offset estimation errors, for which the model was not retrained. The noise has a mean of 0s and a standard deviation of 1/30 s, hence 3 x standard deviations fall within 0.1 s [1]. As shown in Table D, with such a Gaussian noise, the performance drop is minor. Thus, even without retraining, we find our method is performing robustly with respect to uncertain Δt.
>
> *Table D. Performance of AsyncBEV with Δt noise on All Objects*
> | Method   |  NDS (0s) | NDS (0.25s) |NDS (0.5s) |
> |----------|:----:|:----:|:----:|
> | CMT+AsyncBEV (orig., no Δt noise)  | 72.5| 71.5 | 70.0|
> | CMT+AsyncBEV (Infer w. Δt noise)| 72.3 (↓0.2) | 71.4 (↓0.1)| 69.9 (↓0.1)|
>
> ---
>
> [1] Fan, Meng, et al. "Robust sensor fusion against on-vehicle sensor staleness." In *CVPRW 2025*.

---

### Author Response · Authors · 2025-12-02
**Review Summary**

Dear Area Chair,

We sincerely appreciate your efforts in reviewing our submission. For your convenience, we have summarized the reviewers’ feedback along with our corresponding rebuttals. Please refer to the comment cells for further details. In the following discussion, we use the following abbreviations to denote specific reviewers: R1 for Reviewer Vrjt, R2 for Reviewer LiTB, R3 for Reviewer zA9G, and R4 for Reviewer gWRm.

---

### **Strengths:**

+ **\[R1\&R2\] Conceptually intuitive and interpretable**
+ **\[R1, R2, R3, R4\] Generic and lightweight design**
+ **\[R1, R2, R3, R4\] Consistent performance improvement across various asynchronous settings with marginal runtime overhead.**
+ **\[R2, R3 & R4\] Clear Problem formulation and high practical relevance**
+ **\[R3 & R4\] Clearly written and well presented**

---

### **Main Concerns & Corresponding Rebuttals:**

+ **\[R1\] Missing comparison with the Sensor Fallback strategy**
  + We added new experiments for the single-modal performance of both CMT and UniBEV to the Appendix.
  + The results show these baselines underperform our CMT+AsynBEV and UniBEV+AsyncBEV in the most severe asynchronous case, by a large margin.
+ **\[R1 & R3\] Missing comparison with the temporal propagation method Fan et al. \[1\]**
  + We added experiments with Fan et al to Table 1 in Experiments.
  + Fan et al. exhibit a significant performance drop in the synchronous case, but it improves the performance in other asynchronous cases compared to the EMC baseline. However, its performance remains inferior to that of our proposed AsyncBEV in all scenarios.
+ **\[R1 & R4\] Missing comparison with feature prediction method StreamingFlow \[2\]**
  + We highlighted the key difference for feature alignment between StreamingFlow and AsyncBEV.
  + We presented the experiments of StreamingFlow’s two variants. StreamingFlow\* is trained and evaluated using the same input format as AsyncBEV, while StreamingFlow† takes additional historical frames. StreamingFlow† outperforms StreamingFlow\* in both synchronous and asynchronous scenarios. Nevertheless, StreamingFlow† still underperforms our AsyncBEV in both scenarios, and its FPS is much lower than AsyncBEV.
  + We added the experiment of StreamingFlow† to Table 1in Experiments.
+ **\[R1\] Robustness with uncertain Δt**
  + We explained that the timestamp of each individual sensor is reasonably accurate in an autonomous driving system and all our experiments on nuScenes have used real-world sensor timestamps for the Δt calculation, which contains real-world uncertainty.
  + We also evaluated our model on Δt with added Gaussian noise, simulating time offset estimation errors. With such Gaussian noise, the performance drop is minor. Thus, even without retraining, our method performs robustly with respect to uncertain Δt.
+ **\[R2\] Limited novelty compared to collaborative perception methods, CoBEVFlow \[3\] and UniV2X \[4\]**
  + We compared the object proposal-based flow generation technologies in CoBEVFlow and UniV2X with Δ-BEVFlow.
  + We analyzed the shortcomings of these object proposal-based methods when applying them to the asynchronous multi-modal 3D object detection task.
  + We added this discussion to Related Work and highlighted the novelty and value of Δ-BEVFlow in relation to both these prior works.
+ **\[R2 & R4\] Extension for multi-sensor asynchrony**
  + We now explain the multi-sensor asynchrony extension of AsyncBEV in the Appendix.
  + We analysed the time and memory consumption of this extension.
+ **\[R2\] Synchronous performance trade-off**
  + We explained that the performance drop in the synchronous case is very minor compared to the larger performance differences introduced by different architectures.
  + We proposed a different training strategy with varying weights for synchronous and asynchronous cases, which might help address this trade-off.
+ **\[R4\] Performance on another dataset**
  + We admitted that we intended to explore other datasets. Unfortunately, due to the lack of pre-trained models on another dataset, reproducing the results on that dataset would require significant additional effort and we cannot produce the results before the rebuttal deadline.

We believe we have carefully addressed all concerns. We hope this summary helps you in your assessment of our submission.

---

\[1\] Fan, Meng, et al. "Robust sensor fusion against on-vehicle sensor staleness." *In CVPRW* 2025\.
\[2\] Shi, Yining, et al. "Streamingflow: Streaming occupancy forecasting with asynchronous multi-modal data streams via neural ordinary differential equation." In *CVPR* 2024\.
\[3\] Wei, Sizhe, et al. "Asynchrony-robust collaborative perception via bird's eye view flow." *in NeurIPS 2023*.
\[4\] Yu, Haibao, et al. "End-to-end autonomous driving through v2x cooperation." *in* *AAAI*  2025\.

---

### Meta-Review · Area_Chair_whXK · 2025-12-26

**Summary:**

The reviewers' concerns can be summarized as follows:
1. Missing comparison with some methods (Reviewer Vrjt, zA9G, gWRm);
2. Limited Novelty (Reviewer LiTB);
3. Lack of cross-domain and multiple sensors generalization. (Reviewer gWRm);
4. Robustness (Reviewer Vrjt);
5. Synchronous performance trade-off (Reviewer LiTB).

**Reviewer Concerns:**

I think the author's response addresses concerns 1, 4, and 5, but concerns 2 and 3 are not fully resolved.

**Reviewer Scores:**

Given that reviewer Vrjt's concerns have been largely addressed, and considering the comments from other reviewers, reviewer Vrjt is likely to raise the score.

---

### Decision · Program_Chairs · 2026-01-26

Accept (Poster)